# Unleashing the Potential of Vision-Language Pre-Training for 3D Zero-Shot Lesion Segmentation via Mask-Attribute Alignment

**Yankai Jiang**[1*], **Wenhui Lei**[1,2✉], **Xiaofan Zhang**[1,2✉], **Shaoting Zhang**[1✉]

[1]Shanghai AI Laboratory  [2]Shanghai Jiao Tong University

{jiangyankai,zhangshaoting}@pjlab.org.cn, {wenhui.lei,xiaofan.zhang}@sjtu.edu.cn

## Abstract

Recent advancements in medical vision-language pre-training models have driven significant progress in zero-shot disease recognition. However, transferring image-level knowledge to pixel-level tasks, such as lesion segmentation in 3D CT scans, remains a critical challenge. Due to the complexity and variability of pathological visual characteristics, existing methods struggle to align fine-grained lesion features not encountered during training with disease-related textual representations. In this paper, we present **Malenia**, a novel **m**ulti-sc**a**le **le**sio**n**-level mask-attr**i**bute **a**lignment framework, specifically designed for 3D zero-shot lesion segmentation. **Malenia** improves the compatibility between mask representations and their associated elemental attributes, explicitly linking the visual features of unseen lesions with the extensible knowledge learned from previously seen ones. Furthermore, we design a Cross-Modal Knowledge Injection module to enhance both visual and textual features with mutually beneficial information, effectively guiding the generation of segmentation results. Comprehensive experiments across three datasets and 12 lesion categories validate the superior performance of **Malenia**.

## 1 Introduction

3D medical image segmentation has witnessed rapid advancements in recent years (Isensee et al., 2021; Tang et al., 2022; Ye et al., 2023; Liu et al., 2023; Chen et al., 2024; Zhang et al., 2024). However, most tate-of-the-art (SOTA) methods are restricted to a closed-set setting, where they can only predict categories present in the training dataset and typically fail to generalize to unseen disease categories. Given the diversity and prevalence of new anomalies in clinical scenarios, along with the challenges of medical data collection, there is an increasing demand for zero-shot models capable of handling unseen diseases in an open-set setting.

The advent of vision-language pre-training methods, particularly CLIP (Radford et al., 2021), has illuminated a new paradigm for remarkable zero-shot object recognition. This breakthrough also paves the way for significant advancements in zero-shot disease detection and diagnosis. Numerous recent methods (Huang et al., 2021; Tiu et al., 2022; Wu et al., 2023; Phan et al., 2024; Hamamci et al., 2024) align visual and textual features of paired medical image-report data, enabling transferable cross-modal representations. However, leveraging the zero-shot capability of vision-language pre-training for 3D lesion/tumor segmentation remains a scarcely explored area. This extension is nontrivial and faces two obvious challenges: (i) The substantial gap between the upstream contrastive pre-training task and the downstream per-pixel dense prediction task. The former focuses on aligning image-level global representations with text embeddings, while

---

*Work under support from the Shanghai Artificial Intelligence Laboratory. ✉Corresponding authors. Codes are available at https://github.com/Yankai96/Malenia.

the latter requires fine-grained lesion-level visual understanding. This inherent gap necessitates the development of more advanced fine-grained vision-language alignment techniques that can facilitate the perception of nuanced, patient-specific pathological visual clues based on the text descriptions. (ii) Lesions can exhibit significant variations in shape and size, and may present with blurred boundaries. Models often struggle when encountering unseen lesion types due to their out-of-distribution visual characteristics. Simply using text inputs, such as raw reports (Boecking et al., 2022; Tiu et al., 2022; Hamamci et al., 2024), or relying on common knowledge of disease definitions (Wu et al., 2023; Jiang et al., 2024), is insufficient for learning generalized representations needed to segment novel lesions not mentioned in the training dataset.

Motivated by the aforementioned limitations, we introduce **Malenia**, a novel **m**ulti-sc**a**le **le**sio**n**-level mask-attr**i**bute **a**lignment framework for superior zero-shot lesion segmentation. **Malenia** first leverages multi-scale mask representations with inherent boundary information to capture diverse lesion regions, then matches fine-grained visual features of lesions with text embeddings, effectively bridging the gap between the contrastive pre-training task and the per-pixel dense prediction task. To learn extensible representations that are robust to the out-of-distribution visual characteristics of unseen lesions, we incorporate domain knowledge from human experts to structure textual reports into descriptions of various elemental disease visual attributes (*e.g.*, shape, intensity, location). Despite the significant variability among lesions, these fundamental attributes are shared across different diseases and are often represented similarly in images. By aligning mask representations of lesions with their corresponding visual attributes, the model mimics the decision-making process of human experts, explicitly linking the visual features of unseen diseases to the intrinsic attributes learned from seen lesions. Furthermore, we propose a novel Cross-Modal Knowledge Injection (CMKI) module in **Malenia**, inspired by the observation that visual and textual embeddings, after feature alignment, are complementary and can mutually reinforce each other. The CMKI module updates both mask and attribute embeddings to facilitate fine-grained multi-modal feature fusion. We leverage both enhanced mask and attribute embeddings to generate predictions by matching query features with image features, and then ensemble these predictions to produce the final segmentation results, demonstrating improved performance for both seen and unseen lesions. To thoroughly validate the effectiveness of **Malenia**, we evaluate its segmentation performance on both seen and unseen lesions using the MSD (Antonelli et al., 2022), KiTS23 (Heller et al., 2023), and a curated real-world in-house dataset. **Malenia** consistently outperforms state-of-the-art methods across 12 lesion categories. Our contributions can be summarized as follows:

- We present **Malenia**, a novel multi-scale lesion-level mask-attribute alignment framework that captures extensible multi-modal representations for significantly improved zero-shot lesion segmentation by effectively matching the fine-grained visual appearances of new diseases with the textual representations of various fundamental pathological attributes.

- **Malenia** introduces a novel Cross-Modal Knowledge Injection (CMKI) module that enriches both mask and text embeddings with mutually beneficial information through feature fusion, leveraging their complementary strengths to further enhance lesion segmentation performance.

- State-of-the-art lesion segmentation performance in the zero-shot setting across three datasets. **Malenia** significantly outperforms previous methods, and key ablation experiments demonstrate the effectiveness of our strategies in handling lesions with varying characteristics.

## 2 RELATED WORKS

**Medical Vision-Language Pre-training.** Numerous methods (Huang et al., 2021; Tiu et al., 2022; Wu et al., 2023; Phan et al., 2024; Hamamci et al., 2024; Lai et al., 2024) build upon CLIP and align medical images with their corresponding reports or disease definitions to enable zero-shot disease classification. However, these works focus on single body parts (*e.g.*, chest), which limits their applicability in broader medical contexts. Additionally, these methods primarily adhere to a paradigm that matches text embeddings with global image-level or patch-level semantics, originally designed for classification tasks. When adapting their

zero-shot capability to fine-grained segmentation tasks, mismatches can occur due to the model's inability to align text embeddings with detailed pixel features. A recent work, CT-GLIP (Lin et al., 2024) uses organ-level vision-language alignment for zero-shot organ classification and abnormality detection, but due to architectural limitations, it requires fine-tuning with a segmentation head and lacks zero-shot segmentation capabilities. In contrast, we extend the zero-shot capability of vision-language pre-training from image-level to pixel-level by aligning lesion features with fundamental multi-aspect disease visual attributes.

**Zero-Shot 3D Medical Image Segmentation.** Motivated by the impressive zero-shot performance of SAM (Kirillov et al., 2023) and SAM 2 (Ravi et al., 2024) in natural images, numerous studies have evaluated their application in medical image segmentation (Wald et al., 2023; Zhang et al., 2023; Huang et al., 2024b; Yamagishi et al., 2024) and explored effective adaptations of SAM or SAM 2 on medical datasets (Ma et al., 2024; Guo et al., 2024; Shen et al., 2024; Zhu et al., 2024; Shaharabany & Wolf, 2024). Nevertheless, these SAM-based medical image segmentation methods require prompts (points, bounding boxes, or masks) sampled from ground truth during testing, which demands significant expertise and is often impractical in real-world clinical scenarios. As a result, prompt-free SAM adaptation methods (Hu et al., 2023; Zhang & Liu, 2023; Cheng et al., 2024; Aleem et al., 2024) have also been proposed. To the best of our knowledge, although the current SAM-based 3D medical image segmentation methods demonstrate promising zero-shot performance in segmenting certain organs in CT scans, they have **not** been evaluated or proven effective when confronted with unseen lesions that have less defined structures. Apart from SAM-based models, a self-prompted method, ZePT (Jiang et al., 2024), achieves competitive zero-shot tumor segmentation performance by matching class-agnostic mask proposals with text descriptions of general medical knowledge. However, it overlooks patient-specific information from reports, leading to compromised zero-shot performance. Our approach goes a step further, achieving significantly superior zero-shot lesion segmentation performance without the need for complex visual prompts.

**Language-Guided Medical Image Segmentation.** Recently, several methods also incorporate vision and language information to enhance medical image segmentation (Li et al., 2023; Liu et al., 2023; Huang et al., 2024a). Notably, LViT (Li et al., 2023) leverages medical texts to compensate for quality deficiencies in image data and guide the generation of pseudo labels in semi-supervised settings. RecLMIS (Huang et al., 2024a) introduces conditioned reconstruction to explicitly capture cross-modal interactions between medical images and text, leading to enhanced performance. However, these methods primarily focus on fully-supervised and semi-supervised settings, aiming to enhance segmentation performance for seen categories by combining visual and textual features. In contrast, **Malenia** targets the zero-shot setting, aiming to develop generalized and robust textual features, such as disease attributes, along with a novel vision-language alignment strategy to enhance segmentation performance for unseen lesions.

## 3 METHOD

Fig. 1 illustrates the pipeline of **Malenia**. It is built upon the recent mask-based segmentation backbone Mask2Former (Cheng et al., 2022). Below, we introduce (i) our novel multi-scale mask-attribute alignment strategy (Sec.3.1) that effectively aligns lesion representations with text embeddings of fundamental disease attributes; (ii) the Cross-Modal Knowledge Injection module (Sec.3.2) for multi-modal feature representation enhancement; and (iii) the overall training objectives and inference details (Sec. 3.3). We provide the preliminaries, discussing Mask2Former and the problem formulation of zero-shot lesion segmentation, as prior information required for our work in Appendix A.

### 3.1 MULTI-SCALE MASK-TEXT CONTRASTIVE LEARNING

Given a set of $B$ image-report pairs, $\mathbb{D} = \{(\mathcal{I}_1, \mathcal{R}_1), ..., (\mathcal{I}_B, \mathcal{R}_B)\}$, we aim to establish lesion-level alignment between the mask and text representations within the multi-modal feature space. We first adopt a 3D

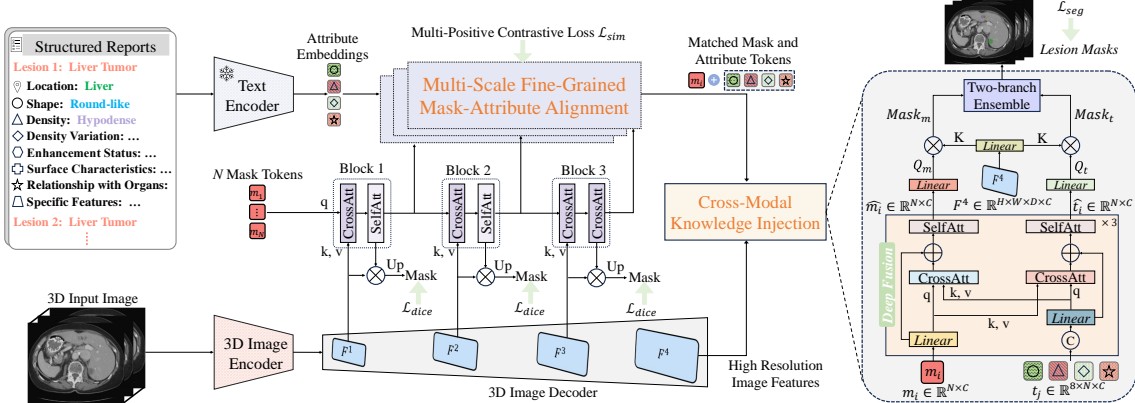

Figure 1: Overview of **Malenia**. The key contributions of our work are two simple but effective designs: the Multi-scale Fine-Grained Mask-Attribute Alignment and the CMKI module.

image encoder to extract high-level features $f$ from the input 3D images $\mathcal{I}$. Then a 3D image decoder gradually upsamples $f$ to generate multi-scale high-resolution per-pixel image features $F^i \in \mathbb{R}^{H^i \times W^i \times D^i \times C^i}$. Here, $H^i$, $W^i$, $D^i$ and $C^i$ denote the height, width, depth and channel dimension of $F^i$, respectively, which vary depending on the resolution level. Subsequently, we feed $N$ learnable mask tokens (queries) $M = \{m_1, \cdots, m_N\}$, along with successive feature maps $F^i$ from the initial three layers of the 3D image decoder into a Transformer decoder with 3 blocks in a round-robin fashion to process the mask tokens. At each block $i \in [1, 3]$, the mask tokens $M$ undergoes a series of layer-wise attention refinements, including cross attention and self-attention as follows:

$$M^{(i)}_{cross} = \text{CrossAttn}(q^{(i)}, k^{(i)}, v^{(i)}), \quad M^{(i)} = \text{SelfAttn}(M^{(i)}_{cross}) \tag{1}$$

Here, $q^{(i)}$, $k^{(i)}$, and $v^{(i)}$ denote the query matrices of mask tokens and the key and value matrices of image features from the 3D image decoder at the $i$-th resolution level, respectively. During the cross-attention phase, mask tokens interact with image features, focusing on the specific regional context within the image. In the self-attention phase, mask tokens interact with each other to enhance the understanding of relationships between different anatomical areas. Building on the segment-level embeddings of mask tokens, we establish fine-grained alignment between mask and text representations through a contrastive learning approach. This vision-language alignment in **Malenia** has three novel key components:

(1) **Utilization of Multi-Scale Features.** Existing methods (Jiang et al., 2024; Lin et al., 2024) overlook the advantage of leveraging multi-scale visual features during cross-modal alignment. In contrast, we match the hierarchical mask token embeddings from different Transformer decoder blocks with text features. Specifically, the mask tokens interact with image features whose dimensions are set as $(h^i, w^i, D^i) = (H/32, W/32, D/32), (H/16, W/16, D/16), (H/8, W/8, D/8)$ for blcoks $i = 1, 2, 3$, respectively. This variation in feature resolution across blocks ensures mask-text alignment at different scales, which is crucial for segmenting classes with large size variations, such as tumors.

(2) **Dissecting Reports into Descriptions of Fundamental Disease Attributes.** Human experts make diagnoses by carefully analyzing key image features that describe distinctive lesion attributes (*e.g.*, shape and density) across different disease classes (Ganeshan et al., 2018; Nobel et al., 2022). Drawing from this inspiration, we consult medical experts and decompose reports into structured descriptions of 8 shared visual attributes of disease imaging characteristics, including **location**, **shape**, **density**, **density variations**, **surface characteristics**, **enhancement status**, **relationship with surrounding organs** and **specific features**.

Specifically, we adopt a semi-automatic pipeline to transform patients' reports into structured descriptions. First, we prompt the Large Language Model (LLM) GPT-4 (Achiam et al., 2023) to extract descriptions related to the lesions from the findings section of each report. Then, two experienced radiologists collaborate to review, correct, supplement, and expand the GPT-generated visual descriptions based on the CT images and corresponding original reports. Leveraging disease attribute descriptions offers two key advantages. First, disease attributes provide fine-grained prior knowledge about the visual manifestations of pathologies, inherently improving alignment with target diseases. Second, new diseases can be described using the elemental visual attributes of previously seen diseases, thereby enhancing the model's zero-shot capability.

(3) **Multi-Positive Contrastive Loss.** Given the multi-scale lesion-level mask embeddings $M$ and visual attribute descriptions, we construct multiple positive and negative pairs, which are then used to learn to optimize a distance metric that brings the positive pairs closer while pushing the negative pairs away. At the $i$-th resolution scale, we obtain $N$ binary mask proposals $BM^{(i)} \in [0,1]^{H^i \times W^i \times D^i \times N}$ by a multiplication operation between the mask embeddings and image features followed by a Sigmoid. We then apply bipartite matching between the upsampled binary mask proposals and the ground truth masks, following (Cheng et al., 2022), to select $S$ foreground mask tokens that correspond to the $S$ lesions in the input 3D CT image in a one-to-one manner. Next, we feed the ground truth descriptions of all visual attributes into the text encoder followed by a MLP layer to acquire the text embeddings $T = \{t_1, \cdots, t_R\}$, where $R$ denotes the number of different attribute descriptions. In our pipeline, we also require a background category for the $N - S$ background mask tokens that do not have a matched ground truth segment. Therefore, we add an extra learnable embedding, $t_0 \in \mathbb{R}^C$, representing "no lesion found". Finally, each foreground mask token is paired with its corresponding eight distinct text features, forming multiple positive sample pairs. While each background mask token is paired with $t_0$. For the $j$-th mask token, let $\mathbb{P}_j = \{k | (m_j, t_k) \text{ is a positive pair}\}$ represent the set of all its positive text embedding indices, and let $\mathbb{N}_j = \{k | (m_j, t_k) \text{ is a negative pair}\}$ represent the set of all its negative text embedding indices. We calculate the pairwise similarity score $S(m_j, t_k)$ between the $j$-th mask token $m_j$ and the $k$-th text embedding $t_k$ as a dot product, normalized by a temperature parameter $\tau$, given by $S(m_j, t_k) = \frac{m_j \cdot t_k}{\tau}$. The lesion-level mask-attribute alignment at the $i$-th resolution level is refined using a contrastive loss function $\mathcal{L}_{sim}^{(i)}$ designed to maximize the similarity scores of positive pairs while minimizing those of negative pairs. Since each lesion has eight positive text embeddings of visual attributes, our framework includes multiple positive pairs for each foreground mask token. The commonly used $N$-pair loss (Sohn, 2016) and InfoNCE loss (Oord et al., 2018), which handle a single positive pair and multiple negative pairs, are not suitable. Therefore, we adopt the Multi-Positive NCE (MP-NCE) loss (Lee et al., 2022) to define the $\mathcal{L}_{sim}^{(i)}$ as:

$$\mathcal{L}_{sim}^{(i)} = -\frac{1}{N} \sum_{j=1}^{N} \mathbb{E}_{k \in \mathbb{P}_j} \left[ \log \frac{\exp\left(S(m_j^{(i)}, t_k)\right)}{\exp\left(S(m_j^{(i)}, t_k)\right) + \sum_{n \in \mathbb{N}_j} \exp\left(S(m_j^{(i)}, t_n)\right)} \right] \tag{2}$$

In this way, we explicitly brings lesion-level mask embeddings closer to their corresponding attribute features while distancing them from unrelated ones. This enables the textual features of each attribute to act as a bridge between the visual features of different diseases, effectively improving the model's zero-shot performance by linking the attributes of unseen lesions with base visual knowledge. Moreover, we apply $\mathcal{L}_{sim}^{(i)}$ to output mask tokens from each Transformer decoder block, utilizing multi-scale feature maps. The multi-scale lesion-level mask-attribute alignment loss is formulated as $\mathcal{L}_{sim} = -\frac{1}{L} \sum_{i=1}^{L} \mathcal{L}_{sim}^{(i)}$, where $L = 3$ denotes the number of Transformer Decoder blocks.

## 3.2 Cross-Modal Knowledge Injection Module

In **Malenia**, we introduce a novel Cross-Modal Knowledge Injection (CMKI) module to generate the final segmentation predictions. This module enriches the features of one modality by incorporating information

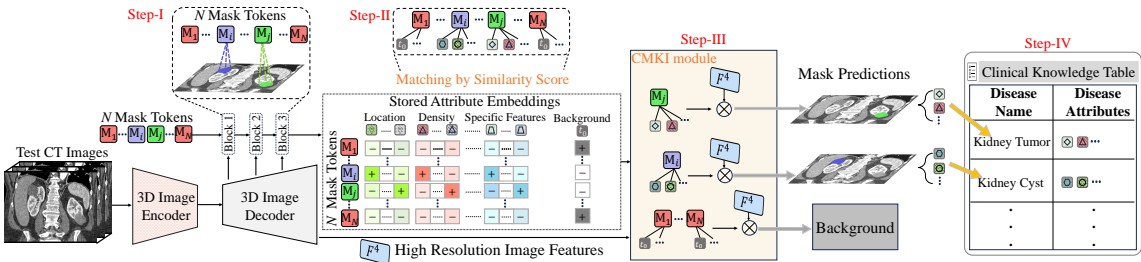

Figure 2: Overview of the inference process of **Malenia**. (1) Step-I: Image Partitioning via Mask Tokens. Test CT images are divided into regions, each represented by mask tokens. (2) Step-II: Mask-attribute matching. Each mask token is associated with stored attribute embeddings. (3) Step-III: Cross-modal fusion and mask prediction. Information from mask tokens and text embeddings is fused to generate segmentation masks. (4) Step-IV: Disease identification via attribute-querying. The Clinical Knowledge Table links the predicted attributes to specific disease categories for precise diagnosis.

from another, enabling deeper understanding and improved feature representations. Specifically, we fuse the output mask tokens $m_i, i \in [1, N]$ from the last Transformer decoder block with their corresponding positive attribute embeddings. As shown in Fig. 1 (a), for each mask token, we first concatenate all its corresponding attribute embeddings $t_j, j \in \mathbb{P}_i$, and transform them into a single text embedding $t_i$ using an MLP layer to obtain the textual features for a comprehensive description of the mask token $m_i$. Then we accomplish deep fusion between two modalities through a series of cross-attention and self-attention layers:

$$\hat{m}_i = \text{SelfAttn}\big(\text{CrossAttn}(q_m, k_t, v_t) + m_i\big), \quad \hat{t}_i = \text{SelfAttn}\big(\text{CrossAttn}(q_t, k_m, v_m) + t_i\big) \qquad (3)$$

Here, $q_m$, $k_m$, and $v_m$ represent the query, key, and value matrices of the mask tokens, while $q_t$, $k_t$, and $v_t$ represent those of the attribute embeddings. The deep fusion of vision and language offers two key benefits: 1) Mask representations are enriched with textual information from language models, resulting in more context-aware segmentation. 2) Text embeddings enhance their descriptive capabilities by attending to visual features, enabling segmentation conditioned on specific text prompts.

We leverage both enhanced mask tokens and text embeddings to generate predictions. By capitalizing on fine-grained vision-language alignment, we formulate semantic segmentation as a matching problem between representative multi-modal query embeddings and pixel-level image features. Given the $N$ enhanced mask tokens $\hat{m} \in \mathbb{R}^{N \times C}$ and text embeddings $\hat{t} \in \mathbb{R}^{N \times C}$, as well as the output pixel-level image features $F^4 \in \mathbb{R}^{H \times W \times D \times C}$ from the last 3D image decoder layer, we apply linear projections $\phi$ to generate $Q$ (query) and $K$ (key) embeddings as:

$$Q_m = \phi_m(\hat{m}) \in \mathbb{R}^{N \times C}, \quad Q_t = \phi_t(\hat{t}) \in \mathbb{R}^{N \times C}, \quad K = \phi_k(F^4) \in \mathbb{R}^{H \times W \times D \times C}. \qquad (4)$$

Then, the mask predictions could be calculated by the scaled dot product attention:

$$Mask_m = \frac{Q_m K^T}{\sqrt{C}} \in \mathbb{R}^{H \times W \times D \times N}, \quad Mask_t = \frac{Q_t K^T}{\sqrt{C}} \in \mathbb{R}^{H \times W \times D \times N} \qquad (5)$$

Here, $Mask_m$ and $Mask_t$ refer to the two-branch output mask predictions derived from mask tokens and text embeddings, respectively. $\sqrt{C}$ is the dimension of the keys as a scaling factor. We ensemble these mask predictions as $Mask = MLP(\beta_1 Mask_m + \beta_2 Mask_t)$. Here $\beta_1$ and $\beta_2$ are weighting factors for the vision and language branches, respectively, set to $0.5$ by default. The final segmentation results are obtained by applying the Argmax operation along the channel dimension of the $Mask$.

### 3.3 TRAINING OBJECTIVES AND INFERENCE

**Overall Loss Function:** We adopt a composite loss function that balances mask-attribute alignment and mask segmentation. Specifically, we use the dice loss $\mathcal{L}_{dice}^{i}$ to supervise the segmentation predictions of mask tokens matched with ground truth at each feature level $i$, achieving deep supervision: $\mathcal{L}_{deep} = -\frac{1}{L} \sum_{i=1}^{L} \mathcal{L}_{dice}^{(i)}$. Additionally, we have the similarity loss $\mathcal{L}_{sim}$ defined by Eq. (2) for aligning mask tokens with attribute embeddings. For the final segmentation results generated by both visual and textual features, we utilize the binary cross-entropy loss and dice loss: $\mathcal{L}_{seg} = \alpha_1 \mathcal{L}_{ce} + \alpha_2 \mathcal{L}_{dice}$. We set $\alpha_1 = 2.0$ and $\alpha_2 = 2.0$. The overall loss function is a weighted sum of these components:

$$\mathcal{L} = \lambda_1 \mathcal{L}_{deep} + \lambda_2 \mathcal{L}_{sim} + \lambda_3 \mathcal{L}_{seg}. \tag{6}$$

where $\lambda_1$, $\lambda_2$, and $\lambda_3$ are weighting factors balancing the contribution of each loss component to the overall training objective. We set $\lambda_1 = \lambda_2 = \lambda_3 = 1.0$ as default.

**Inference:** Fig. 2 illustrates the model testing flow. To enable convenient and flexible inference, we store the text features of all visual attribute descriptions after training, eliminating the need for the text encoder during testing. The inference process can be divided into four steps. **Step-I**: Mask tokens attend to different regions among the images, partitioning the image into different regions. In this process, different lesion regions in the image are captured by specific mask tokens. **Step-II**: We compute the similarity between the $N$ mask tokens and the stored attribute embeddings, after which the mask tokens are matched to the text embeddings based on these similarity scores. **Step-III**: The matched mask tokens and text tokens are fed into the CMKI module for further feature fusion. Then, the information from mask tokens and text embeddings are combined to produce the final mask prediction. **Step-IV**: The text attributes of each mask are used to identify the specific category. This is achieved by querying a Clinical Knowledge Table that records the relationships between each disease and its attributes. The Clinical Knowledge Table are given in Table 11.

## 4 EXPERIMENTS

### 4.1 EXPERIMENT SETUP

**Datasets.** We utilize annotated datasets encompassing lesions from 12 diseases and 6 organs, sourced from both public and private datasets. (1) Public datasets: We consider the MSD dataset (Antonelli et al., 2022) and KiTS23 (Heller et al., 2023) dataset. Two senior radiologists supplemented the annotated lesions in MSD and KiTS23 datasets with diagnostic reports and corresponding lesion descriptions of the eight visual attribute aspects. We also collect a private dataset[*] that includes four lesion types: liver cyst, gallbladder tumor, gallstone, and kidney stone. For *training*, we use the following datasets: 1) colon tumor, lung tumor, liver tumor, and pancreas tumor from the MSD dataset; 2) kidney cyst from the KiTS23 dataset; 3) gallstone from our private dataset. For *zero-shot testing*, we adopt: 1) hepatic vessel tumor and pancreas cyst from the MSD dataset; 2) kidney tumor from the KiTS23 dataset; and 3) liver cyst, gallbladder tumor and kidney stone from our private dataset. Details of all datasets, the annotation process for disease attribute descriptions, and preprocessing are provided in Appendix B.

**Evaluation Metrics.** We adopt standard segmentation metrics, including the Dice Similarity Coefficient (DSC) and Normalized Surface Distance (NSD). Additionally, we report the the computational efficiency evaluation, which are detailed in Appendix G.

**Implementation Details.** We use nnUNet (Isensee et al., 2021) as the backbone for 3D image encoder and 3D image decoder. We choose Clinical-Bert (Alsentzer et al., 2019) as the text encoder. We employ AdamW

---

[*]This retrospective study was approved by the ethics committee of the First Hospital of China Medical University; the informed consent requirement was waived. All diagnostic reports in the private dataset used in this study have been fully anonymized, with all patient-identifiable information and private details removed.

Table 1: Segmentation performance (%) of unseen lesions on the MSD, KiTS23, and our in-house dataset. † denotes methods that adopt SAM or SAM2 for zero-shot medical image segmentation, and ∗ indicates methods implemented using the official code. All competing methods are trained on the same dataset. The best performance is highlighted in light blue.

| Method | MSD | | | | KiTS23 | | In-house Dataset | | | | | |
|---|---|---|---|---|---|---|---|---|---|---|---|---|
| | Hepatic Vessel Tumor | | Pancreas Cyst | | Kidney Tumor | | Liver Cyst | | Kidney Stone | | Gallbladder Tumor | |
| | DSC↑ | NSD↑ | DSC↑ | NSD↑ | DSC↑ | NSD↑ | DSC↑ | NSD↑ | DSC↑ | NSD↑ | DSC↑ | NSD↑ |
| SAM† (Shaharabany & Wolf, 2024) | 35.76 | 45.83 | 37.17 | 49.26 | 35.45 | 41.33 | 34.99 | 40.88 | 24.14 | 31.92 | 28.08 | 36.38 |
| SAM2† (Yamagishi et al., 2024) | 35.93 | 45.88 | 38.42 | 50.85 | 35.67 | 41.88 | 35.29 | 41.25 | 25.50 | 33.74 | 28.57 | 36.62 |
| SaLIP∗ (Aleem et al., 2024) | 39.65 | 48.71 | 41.92 | 53.06 | 38.64 | 44.91 | 37.71 | 44.26 | 27.24 | 36.61 | 30.84 | 38.97 |
| H-SAM∗ (Cheng et al., 2024) | 45.58 | 54.24 | 46.87 | 57.91 | 44.21 | 50.39 | 43.75 | 50.20 | 29.23 | 38.11 | 32.17 | 40.05 |
| ZePT∗ (Jiang et al., 2024) | 53.12 | 63.25 | 53.35 | 63.50 | 46.82 | 52.44 | 51.64 | 57.36 | 33.97 | 42.42 | 35.48 | 43.23 |
| **Malenia** | **59.52** | **69.60** | **60.91** | **70.28** | **54.96** | **60.60** | **61.85** | **70.93** | **43.05** | **52.95** | **47.35** | **55.79** |

optimizer (Loshchilov & Hutter, 2017) with a warm-up cosine scheduler of 40 epochs. The batch size is set to 2 per GPU. Each input volume is cropped into patches with a size of $96 \times 96 \times 96$. The training process uses an initial learning rate of $1e^{-4}$, momentum of 0.9, and weight decay of $1e^{-5}$, running on 8 NVIDIA A100 GPUs with DDP for 4000 epochs. The number of the mask tokens $N$ is set as 16.

## 4.2 MAIN RESULTS

**Zero-shot segmentation performance on unseen lesions.** Table 1 presents the zero-shot lesion segmentation results on unseen datasets from various institutions, encompassing a wide range of lesion types. All available CT volumes in these tasks are directly used for testing. We compare **Malenia** with five state-of-the-art zero-shot medical image segmentation methods: the self-prompted ZePT (Jiang et al., 2024), the prompt-free SAM-based method H-SAM (Cheng et al., 2024), the fine-tuning-free method combining CLIP and SAM, SaLIP (Aleem et al., 2024), and two methods (Shaharabany & Wolf, 2024; Yamagishi et al., 2024) that respectively adopt SAM and SAM 2. **Malenia** achieves superior performance across all the datasets, substantially outperforming the other five methods. The weak performance of SAM (Shaharabany & Wolf, 2024) and SAM 2 (Yamagishi et al., 2024) can be attributed to the inherent variability of segmentation tasks in different clinical scenarios. Without target data for fine-tuning and accurate manual prompts, the SAM and SAM 2 struggle with lesion segmentation in the zero-shot setting. Additionally, creating accurate prompts requires domain knowledge from medical experts, which is often limited and time-consuming. SaLIP leverages the capabilities of CLIP to refine automatically generated prompts for SAM. However, its performance is still constrained by CLIP, which focuses on aligning image-level global features while neglecting fine-grained local lesion semantics, resulting in suboptimal performance for lesion segmentation. H-SAM achieves better results by incorporating enhanced mask attention mechanisms in a two-stage decoding process. However, it does not explore cross-modal feature representations. Without the rich textual knowledge from language models, its zero-shot segmentation performance is significantly hindered. ZePT is trained using both organ and lesion labels to generate anomaly score maps as prompts, and it aligns mask features with common medical knowledge to enhance zero-shot tumor segmentation. However, it overlooks the patient-specific multi-aspect elemental attributes shared across different diseases, which limits the scalability and generalization of its learned representations. In contrast, **Malenia** learns fine-grained lesion-level mask-attribute alignment to link unseen lesions with base knowledge and leverages both visual and textual context for advanced cross-modal understanding, resulting in at least a $6.40\%$ improvement in DSC on MSD, a $8.14\%$ improvement in DSC on KiTS23, and a $9.08\%$ improvement in DSC on the in-house dataset.

**Segmentation performance on seen lesions.** We also evaluate the segmentation performance of **Malenia** on seen lesions in the fully-supervised setting. We compare **Malenia** with SOTA medical image segmentation methods, including TransUNet (Chen et al., 2021), nnUNet (Isensee et al., 2021), Swin UNETR (Tang et al., 2022), and the Universal Model (Liu et al., 2023). As shown in Table 2, **Malenia** outperforms competing

Table 2: Segmentation performance (%) of seen lesions on the MSD and KiTS23 dataset. We compare **Malenia** with SOTA supervised methods and report the 5-fold cross-validation results. ∗ means implemented from the official code and trained on the same dataset. The best performance is highlighted in light blue.

| Method | MSD | | | | | | | | KiTS23 | |
| | Colon Tumor | | Pancreas Tumor | | Liver Tumor | | Lung Tumor | | Kidney Cyst | |
| | DSC↑ | NSD↑ | DSC↑ | NSD↑ | DSC↑ | NSD↑ | DSC↑ | NSD↑ | DSC↑ | NSD↑ |
|---|---|---|---|---|---|---|---|---|---|---|
| TransUNet∗ | 44.78±16.21 | 54.14±15.67 | 38.85±10.25 | 54.72±11.59 | 60.05±5.29 | 72.88±5.98 | 67.13±6.08 | 68.89±7.22 | 48.43±14.04 | 52.32±15.62 |
| nnUNet∗ | 47.02±15.85 | 57.36±14.33 | 37.97±10.54 | 53.98±11.86 | 61.33±5.01 | 73.27±5.44 | 69.50±5.61 | 71.39±6.55 | 48.76±13.82 | 52.96±15.19 |
| Swin UNETR∗ | 46.87±16.02 | 55.28±15.52 | 38.72±10.33 | 54.01±11.67 | 62.37±4.88 | 74.75±5.09 | 68.95±5.67 | 71.03±6.82 | 48.06±14.26 | 52.11±16.05 |
| Universal Model∗ | 51.02±14.62 | 60.93±13.36 | 42.40±9.54 | 58.54±10.79 | 64.25±3.94 | 77.06±4.21 | 67.27±5.71 | 69.33±6.95 | 50.25±12.24 | 54.17±13.53 |
| **Malenia** | **53.55±13.49** | **62.41±12.81** | **43.30±9.29** | **59.63±10.55** | **65.18±3.74** | **78.95±4.03** | **70.96±5.56** | **72.34±6.29** | **51.60±11.84** | **55.41±12.99** |

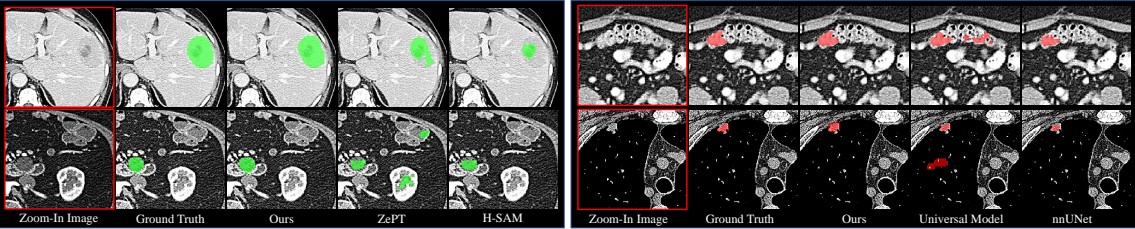

Figure 3: Qualitative visualizations of **Malenia** and other competing methods on both unseen and seen lesions. The segmentation results, presented from top to bottom and left to right, include Hepatic Vessel Tumor, Pancreas Cyst, Colon Tumor, and Lung Tumor.

baselines on seen lesions, achieving an average improvement of 1.46% in DSC for tumors on the MSD dataset (Antonelli et al., 2022) and 1.35% in DSC for kidney cysts on KiTS23 (Heller et al., 2023). Most segmentation baselines (*e.g.*, TransUNet, nnUNet, and Swin UNETR) focus solely on visual features and overlook the semantic relationships between different lesions. Although the Universal Model incorporates text embeddings of category names to learn correlations between anatomical regions, it lacks fine-grained information from both the vision and language domains. In contrast, **Malenia** outperforms these methods by aligning fine-grained lesion-level semantics with comprehensive, patient-specific textual features and enhancing mask representations through cross-modal knowledge injection. This improvement demonstrates that our novel strategies also enhance the segmentation of seen lesion categories.

**Qualitative Comparison.** As shown in Fig. 3, **Malenia** accurately segments various types of lesions across diverse organs, having substantially better performance in segmenting both seen and unseen lesions than the other methods. Most competing methods suffer from incomplete segmentation and false positives. In contrast, **Malenia** produces results that are more consistent with the ground truth. This further demonstrates the superior lesion segmentation capability of **Malenia** on datasets with highly diverse lesion semantics.

### 4.3 ABLATION STUDIES

**Multi-scale mask-attribute alignment strategy ablation.** We validate the effectiveness of different components of the multi-scale mask-attribute alignment strategy on both seen and unseen lesion datasets, as detailed in Table 3. 'Baseline' refers to the naive single-scale mask-report alignment performed at the last Transformer decoder block. We gradually enhance the baseline by ($S_1$) enriching raw reports with structured eight visual attribute descriptions of the lesions; ($S_2$) utilizing multi-scale features for cross-modal alignment; ($S_3$) formulating mask-attribute alignment as multi-positive contrastive learning using multiple mask-attribute positive pairs, rather than a single mask-report positive pair. Each component contributes to the remarkable segmentation performance of **Malenia**. Enriching raw reports with visual attributes of

Table 3: Ablation study of the multi-scale mask-attribute alignment strategy.

| Module | MSD (seen) | | KiTS23 (unseen) | | In-house Dataset (unseen) | |
|---|---|---|---|---|---|---|
| | Pancreas Tumor | | Kidney Tumor | | Gallbladder Tumor | |
| | DSC↑ | NSD↑ | DSC↑ | NSD↑ | DSC↑ | NSD↑ |
| Baseline | 38.77±10.30 | 54.28±11.63 | 47.05 | 53.47 | 36.12 | 43.81 |
| $+S_1$ | 39.83±10.13 | 55.66±11.47 | 48.69 | 54.82 | 38.36 | 46.20 |
| $+S_2$ | 40.46±9.95 | 56.52±11.18 | 49.63 | 55.99 | 39.74 | 47.79 |
| $+S_1+S_2$ | 41.88±9.88 | 57.79±11.05 | 51.85 | 57.88 | 42.53 | 50.67 |
| $+S_1+S_3$ | 42.23±9.71 | 58.24±10.96 | 53.54 | 59.46 | 44.40 | 52.91 |
| $+S_1+S_2+S_3$ | **43.30±9.29** | **59.63±10.55** | **54.96** | **60.60** | **47.35** | **55.79** |

Table 4: Ablation study of the Cross-Modal Knowledge Injection module.

| Module | | | MSD (seen) | | KiTS23 (unseen) | | In-house Dataset (unseen) | |
|---|---|---|---|---|---|---|---|---|
| | | | Pancreas Tumor | | Kidney Tumor | | Gallbladder Tumor | |
| TE | MT | DF | DSC↑ | NSD↑ | DSC↑ | NSD↑ | DSC↑ | NSD↑ |
| ✓ | | | 38.96±10.22 | 54.79±11.52 | 47.24 | 53.68 | 36.38 | 44.02 |
| | ✓ | | 40.27±10.01 | 56.34±11.25 | 49.09 | 55.14 | 39.65 | 47.42 |
| ✓ | ✓ | | 41.22±9.90 | 57.15±11.10 | 52.58 | 58.50 | 43.07 | 51.23 |
| ✓ | | ✓ | 40.50±9.93 | 56.61±11.14 | 51.77 | 57.83 | 41.28 | 49.19 |
| | ✓ | ✓ | 42.44±9.48 | 58.59±10.70 | 52.87 | 58.63 | 43.69 | 51.95 |
| ✓ | ✓ | ✓ | **43.30±9.29** | **59.63±10.55** | **54.96** | **60.60** | **47.35** | **55.79** |

lesions ($S_1$) helps the model leverage pre-established knowledge of the diseases' visual manifestations to enhance the alignment of fine-grained image features with the representations of target diseases, thereby improving segmentation performance. Multi-scale cross-modal alignment ($S_2$) leverages multi-level features to accurately capture and segment both seen and unseen lesions across various sizes, which is essential for handling the shape and size variations of lesions. Furthermore, instead of combining the eight visual attribute descriptions of each lesion into a single paragraph and then extracting the text features ($S_1$), we extract the text features for each visual attribute description separately ($S_3$). As a result, the mask embeddings for each lesion is paired with eight distinct text features, forming multiple positive sample pairs. Simply treating a lesion's attribute descriptions as a single paragraph yields only one positive sample pair for each foreground mask token. Consequently, reports of other lesions that share some of the same attribute descriptions are treated as negative samples, leading to compromised feature representations. In contrast, our formulation of the multi-positive contrastive learning ($S_3$) focuses on establishing comprehensive and extensible correlations between pathological features and fundamental disease attributes. This enables the model to associate the visual cues of unseen lesions with foundational visual knowledge, allowing for effective segmentation of new diseases by translating their complex visual features into elemental attributes shared with seen diseases. This significantly enhances zero-shot lesion segmentation performance.

**Ablation of the Cross-Modal Knowledge Injection module.** We examine key components in the proposed CMKI module in detail, including (1) the significance of deep fusion (DF), which is designed to enhance the representation of mask tokens and text embeddings; (2) the effectiveness of leveraging both text embeddings (TE) and mask tokens (MT) to generate predictions. The results are shown in Table 4. Comparing the results of the first three rows with the last three rows (highlighted in light red), it is evident that deep fusion significantly improves performance, whether using only text embeddings, only mask tokens, or both for segmentation result prediction. This observation shows the importance of enabling cross-modal information interaction. Furthermore, whether or not deep fusion is applied, using both text embeddings and mask tokens for segmentation prediction, combined with ensembling the results, consistently outperforms using only unimodal token embeddings for mask prediction. This confirms the superiority of leveraging the complementary strengths of both visual and textual embeddings to further enhance segmentation performance.

## 5 CONCLUSION

In this work, we propose **Malenia**, a novel vision-language pre-training method designed for 3D zero-shot lesion segmentation, which incorporates an innovative multi-scale lesion-level vision-language alignment strategy. Inspired by the image interpretation process of human experts, we transform patient-specific reports into structured descriptions of disease visual attributes and then match them with mask representations. Additionally, we introduce a novel Cross-Modal Knowledge Injection module, which enhances cross-modal representations and leverages the complementary strengths of both visual and textual features to further improve segmentation performance. Extensive experiments demonstrate that **Malenia** consistently outperforms previous state-of-the-art approaches across diverse datasets for 3D zero-shot lesion segmentation. We hope this work inspires further innovation in this challenging yet promising research area.

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

# A   PRELIMINARIES

**Problem Formulation of Zero-Shot Lesion Segmentation.** Traditional semantic segmentation is inherently limited to a closed-set setting and often struggles in real-world applications where there can be anomalous categories in the test data unseen during training. In contrast, the goal of zero-shot segmentation is to segment objects belonging to categories that have not been encountered during training. In this work, we explore the zero-shot setting for lesion segmentation in 3D CT images. Specifically, there are two non-overlapping foreground sets: (1) $N^s$ seen (base) categories of lesions denoted as $C^s$; (2) $N^u$ unseen (novel) categories of lesions denoted as $C^u$, $C^s \cap C^u = \varnothing$. The training data is constructed from the images and labels that contain any of the $N^s$ seen categories. Among these images, there may also exist unseen categories from $C^u$, whose annotations can **not** be accessed during the training. **Malenia** aims to segment lesions, both seen and unseen, in the test data.

**Mask Tokens.** Recently, a transformer-based segmentation model Mask2Former (Cheng et al., 2022) that can generate a set of segment-level embeddings have been applied and achieved promising performance in the field of medical image segmentation (Yuan et al., 2023; Chen et al., 2023; Jiang et al., 2024). Different from standard segmentation methods, Mask2Former adopt an additional lightweight transformer decoder along with a vision encoder and a pixel decoder. By feeding $N$ mask tokens (also called segment queries) and pixel decoder features into the transformer decoder, Mask2Former generates $N$ segment-level embeddings with masked attention layers, effectively partitioning an image into $N$ corresponding regions. The final binary mask segmentations are obtained by computing the dot product of the mask tokens with the image features from the last pixel decoder layer, while an additional MLP predicts the class for each mask. In this work, we extend the closed-set setting of mask classification to an open-set setting through fine-grained mask-text alignment. Specifically, we propose enhancing and matching mask tokens with text embeddings of the visual attributes of pathologies within multi-scale transformer decoder blocks. Additionally, we leverage the complementary strengths of visual and text representations for mask refinement, demonstrating strong generalizability and improved zero-shot lesion segmentation performance.

# B   DATASET DETAILS AND DATA PREPROCESSING

## B.1   DATASET DETAILS

Our study utilizes annotated datasets encompassing lesions across 12 diseases and 6 organs, derived from both public and private sources. The private datasets particularly include paired CT scans and radiological reports providing systematic imaging descriptions of lesions. We summarize all the datasets in Table 5.

**Public Datasets:**

- Kidney Tumor and Kidney Cyst. This dataset is part of the Kidney and Kidney Tumor Segmentation Challenge (KiTS23) (Heller et al., 2023), which provides 489 cases of data with annotations for the segmentation of kidneys, renal tumors, and cysts.
- Liver Tumor, Pancreas Tumor, Pancreas Cyst, Colon Tumor, and Lung Tumor. These datasets are part of the Medical Segmentation Decathlon (MSD) (Antonelli et al., 2022), providing annotated datasets for various lesions. Notably, Pancreas Tumors and Cysts are grouped under a single region of interest in the MSD dataset but were separately classified by experienced radiologists as part of this study.

**Private Datasets:**

- For lesions underrepresented in public datasets, we utilized private datasets annotated by radiologists. These datasets include lesions such as kidney stones, liver cysts, gallbladder cancer, and

Table 5: Details of Datasets. ∗ denotes MSD-Pancreas is reannotated to pancreas tumor and pancreas cyst.

| Dataset Name | Segmentation Lesion Targets | # of scans |
|---|---|---|
| KiTS23 | Kidney Tumor, Kidney Cyst | 489 |
| MSD-Colon Tumor | Colon Tumor | 126 |
| MSD-Liver Tumor | Liver Tumor | 131 |
| MSD-Hepatic Vessel Tumor | Hepatic Vessel Tumor | 303 |
| MSD-Lung Tumor | Lung Tumor | 64 |
| MSD-Pancreas∗ | Pancreas Tumor | 216 |
| MSD-Pancreas∗ | Pancreas Cyst | 65 |
| Private Data | Liver Cyst | 30 |
| | Gallbladder Tumor | 30 |
| | Gallstones | 30 |
| | Kidney Stone | 30 |

gallstones, accompanied by paired radiological reports. Each dataset consists of 30 cases, providing a balanced sample size for preliminary analysis.

## B.2 Creating Structured Disease Attribute Descriptions

Specifically, For the public datasets MSD and KiTS23, which only provide mask annotations, two senior radiologists supplemented the annotated lesions with diagnostic reports and corresponding lesion descriptions of the eight visual attribute aspects. For the in-house data, we utilized the semi-automatic annotation pipeline described in Sec.3.1. First, we prompted GPT-4 (Achiam et al., 2023) to extract lesion-related descriptions from the findings section of each report. Then, two experienced radiologists collaborated to review, correct, supplement, and expand the GPT-generated visual attribute descriptions. Through this process, we established eight standardized attributes for describing lesions, which form the basis of our structured reports. These attributes provide a comprehensive and objective template for characterizing lesions, ensuring consistency and enabling generalization across different types of lesions, as shown in Table 6.

## B.3 Data Preprocessing

We pre-process CT scans using isotropic spacing and uniformed intensity scale to reduce the domain gap among various datasets. We determine whether each CT scan is contrast-enhanced by evaluating the Hounsfield unit (HU) values of the aorta and inferior vena cava. If the average HU of both the aorta and inferior vena cava is less than 80, the scan is classified as non-contrast. Otherwise, it is classified as contrast-enhanced. This helps radiologists determine the attribute description "Enhancement Status".

## B.4 Justification for the Legality of Data Use

This study utilizes data from both public and private sources. (1) The public data is obtained from the MSD (Antonelli et al., 2022) and KiTS23 (Heller et al., 2023) datasets, which are publicly available and provide CT images along with lesion masks. We have supplemented these lesion masks with corresponding structured lesion attribute descriptions. Since these datasets are fully open and no patient privacy information was disclosed during their use, our data usage is fully justifiable and legal. (2) The private dataset is sourced from the First Hospital of China Medical University. This retrospective study was approved by the Ethics Committee of the First Hospital of China Medical University, and the informed consent requirement was waived. The private dataset includes CT images, lesion masks, and diagnostic reports. All diagnostic reports in this dataset have been fully anonymized, with all patient-identifiable information and private details re-

Table 6: Definitions and detailed content of the eight disease attributes.

| Attribute | Definition | Content |
|---|---|---|
| Enhancement Status | Intravenous contrast agent usage | "Enhanced CT","Non-contrast CT" |
| Location | Organ-specific anatomical regions | "Colon", "Liver", "Pancreas", "Right Lung", "Left Lung" "Right Kidney", "Left Kidney", "Gallbladder", "Hepatic Vessel" |
| Shape | Morphological characteristics of the lesion | "Round-like", "Irregular", "Wall thickening", "Punctate", "Nodular", "Cystic", "Luminal narrowing", "Protrusion into the lumen" |
| Relationship with Surrounding Organs | Invasion of or proximity to adjacent organs | "No close relationship with surrounding organs", "Close relationship with adjacent organs" |
| Density | Radiographic attenuation properties of the lesion | "Hypodense lesion", "Isodense lesion", "Hyperdense lesion", "Mixed-density lesion", "Hypodense fluid-like lesion", "Isodense soft tissue mass", "Low-density ground-glass opacity" |
| Density Variations | Uniformity of attenuation within the lesion | "Homogeneous", "Heterogeneous" |
| Surface Characteristics | Features of the lesion's border and surface texture | "Well-defined margin", "Clear serosal surface", "Ill-defined margin", "Serosal surface irregularity" |
| Specific Features | Distinctive attributes indicating lesion characteristics | "Spiculated margins", "Retention of pancreatic fluid", ... |

moved. For all anonymized diagnostic reports, we exclusively utilized the GPT-4 API (Achiam et al., 2023) to process textual descriptions of lesions, converting them into a structured format. No patient-identifiable information was disclosed throughout the data processing, ensuring that our data usage is completely justifiable and legal.

## C  FINE-TUNING EVALUATION

**Malenia** can also serve as a valuable pre-training method for downstream segmentation tasks in a fine-tuning setting. To evaluate this, we adopt various vision-language pre-training strategies to pre-train the 3D image encoder of a nnUNet model using our training data. We then fine-tune the nnUNet on downstream datasets using the pre-trained weights. For all downstream datasets, we split each into $50\%$ for training, $20\%$ for validation, and $30\%$ for testing. As shown in Table 7, **Malenia** consistently outperforms previous methods, further validating the importance of tailoring vision-language pre-training for dense prediction tasks in the medical domain, where a fine-grained understanding of disease-related features is essential. Notably, **Malenia** significantly surpasses the second-best method, CT-GLIP, by $6.03\%$ and $3.77\%$ in DSC on hepatic vessel tumor and pancreas cyst segmentation, respectively. On our in-house dataset, **Malenia** also delivers a substantial performance boost. The fine-tuning evaluations demonstrate that injecting fine-grained, lesion-level cross-modal knowledge from both the vision and language domains to enhance mask representations during pre-training improves the model's transferability, particularly for fine-grained lesion segmentation tasks.

Table 7: Segmentation results of different vision-language pretraining models under the fine-tuning setting.

| Method | MSD | | | | KiTS23 | | In-house Dataset | | | | | |
|---|---|---|---|---|---|---|---|---|---|---|---|---|
| | Hepatic Vessel Tumor | | Pancreas Cyst | | Kidney Tumor | | Liver Cyst | | Kidney Stone | | Gallbladder Tumor | |
| | DSC↑ | NSD↑ | DSC↑ | NSD↑ | DSC↑ | NSD↑ | DSC↑ | NSD↑ | DSC↑ | NSD↑ | DSC↑ | NSD↑ |
| nnUNet (from Scratch) | 62.44 | 72.87 | 64.15 | 76.27 | 70.59 | 76.79 | 67.84 | 74.03 | 50.47 | 59.91 | 56.70 | 64.68 |
| GLoRIA (Huang et al., 2021) | 63.11 | 73.02 | 64.55 | 76.41 | 72.03 | 77.82 | 68.35 | 74.87 | 50.82 | 60.25 | 57.53 | 65.46 |
| BioViL (Boecking et al., 2022) | 63.14 | 73.08 | 64.57 | 76.44 | 72.10 | 77.94 | 68.66 | 75.02 | 50.88 | 60.27 | 57.62 | 65.69 |
| MedKLIP (Wu et al., 2023) | 64.30 | 74.45 | 65.20 | 77.35 | 72.87 | 78.71 | 69.15 | 75.56 | 51.42 | 61.65 | 58.34 | 66.40 |
| MAVL (Phan et al., 2024) | 65.49 | 75.51 | 65.49 | 77.61 | 74.09 | 80.23 | 69.23 | 75.60 | 52.30 | 62.14 | 58.71 | 66.77 |
| CT-CLIP (Hamamci et al., 2024) | 65.66 | 75.73 | 65.85 | 77.92 | 74.12 | 80.24 | 69.57 | 75.82 | 52.45 | 62.37 | 58.80 | 66.84 |
| CT-GLIP (Lin et al., 2024) | 65.93 | 75.97 | 66.18 | 78.06 | 74.15 | 80.26 | 69.92 | 76.11 | 52.67 | 62.52 | 58.93 | 66.95 |
| **Malenia** | **71.96** | **81.83** | **69.95** | **82.00** | **76.88** | **82.55** | **72.44** | **79.84** | **53.35** | **63.69** | **64.21** | **72.63** |

Table 8: Ablation study on using different LLMs for attribute construction. As described in Sec. B.2, the LLMs are used only for our in-house dataset, as public datasets lack diagnostic reports and instead rely on attributes directly annotated by human experts.

| Method | In-house Dataset | | | | | |
|---|---|---|---|---|---|---|
| | Liver Cyst | | Kidney Stone | | Gallbladder Tumor | |
| | DSC↑ | NSD↑ | DSC↑ | NSD↑ | DSC↑ | NSD↑ |
| MMed-Llama-3-8B | 54.97 | 63.06 | 38.66 | 47.71 | 43.28 | 51.45 |
| GPT4 | 54.98 | 63.08 | 38.67 | 47.73 | 43.28 | 51.46 |
| MMed-Llama-3-8B + Human Annotators | 61.84 | **70.93** | 43.03 | 52.94 | 47.34 | **55.79** |
| GPT4 + Human Annotators | **61.85** | **70.93** | **43.05** | **52.95** | **47.35** | **55.79** |

# D ADDITIONAL ABLATION STUDIES

## D.1 DIFFERENT LLMS FOR ATTRIBUTE CONSTRUCTION.

To transform patient reports into structured descriptions of fundamental disease attributes, we employ a semi-automatic pipeline that uses GPT-4 (Achiam et al., 2023) for extracting lesion-related descriptions from each report, followed by further refinement by experienced human annotators. Given that using GPT-4 (Achiam et al., 2023) for attribute construction from radiological reports is often impractical due to privacy and legal concerns, we conduct extensive experiments to determine whether switching to locally hosted open-source LLMs can achieve comparable performance. The results are shown in Table 8. We compare the segmentation performance on unseen lesion categories using either the open-source medical large language model MMed-Llama-3-8B (Qiu et al., 2024) or GPT-4 (Achiam et al., 2023) to extract attribute descriptions from radiological reports.

The results indicate that using GPT-4 (Achiam et al., 2023) or MMed-Llama-3-8B (Qiu et al., 2024) for attribute extraction from radiological reports, with or without human annotation, has a negligible effect on model performance. This is largely because extracting and structuring attributes already present in the reports is a straightforward and easy task. However, incorporating human annotators significantly enhances performance, as some attributes are not explicitly mentioned in the reports and require expert supplementation—a task beyond the capabilities of current large language models. Given the high cost of human annotation, and to support future research, we will open-source our attribute annotations for all public datasets used in this study, along with our code, to benefit the research community.

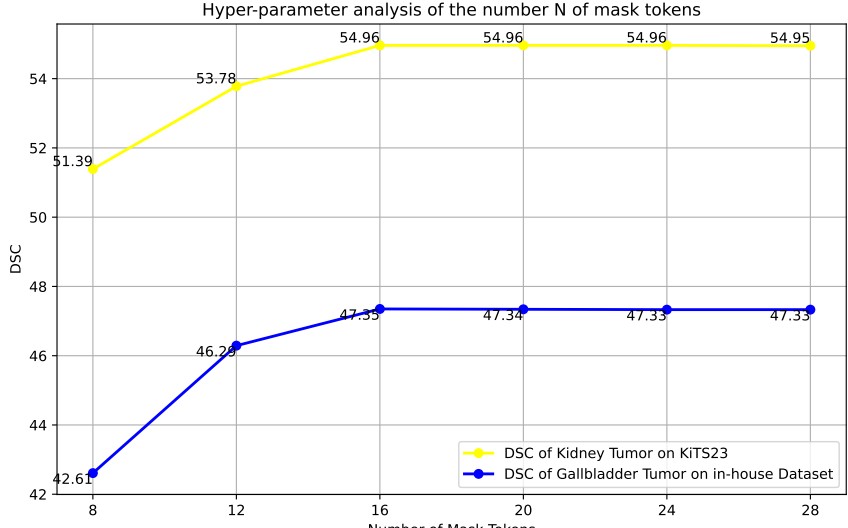

Figure 4: Ablation studies on segmentation performance of unseen lesions using different numbers of mask tokens.

## D.2 THE NUMBER OF MASK TOKENS

An important hyperparameter in our framework is the number of mask tokens. We conducted ablation experiments to assess the segmentation performance of unseen lesions using different numbers of mask tokens on the KiTS23 dataset (Heller et al., 2023) and our in-house dataset. The results are presented in Fig. 4. We observed that when $N$ is less than 12, the model's performance drops sharply. This observation is consistent with previous findings from MaskFormer-based methods (Cheng et al., 2021; 2022; Jiang et al., 2024), which suggest that the number of queries should be greater than the number of possible or useful classes in the data. When $N$ exceeds 16, no significant performance improvement was observed. Therefore, we set the default number of mask tokens to $N = 16$.

## D.3 THE NUMBER OF ATTRIBUTE ASPECTS

This section addresses the key question of whether adding more attribute aspects improves zero-shot segmentation performance. We conducted ablation experiments to evaluate the segmentation performance of unseen lesions as we gradually added attribute categories in the order defined in Table 6. The results are presented in Fig. 5. We observed that as the number of attribute categories used for vision-language alignment increased from 1 to 8, the model's zero-shot performance consistently improved with greater attribute diversity. This result demonstrates the superiority of our approach in using multiple attribute aspects to describe lesions from different perspectives, providing valuable textual features that help capture subtle visual characteristics of the lesions. This finding also suggests that incorporating additional contextual information or advanced features enhance the model's ability to capture complex visual semantics.

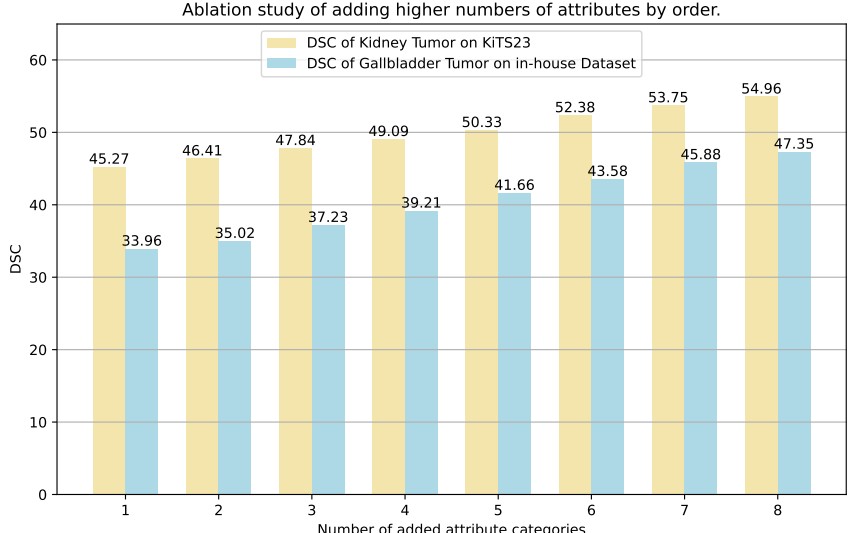

Figure 5: Ablation studies on segmentation performance of unseen lesions using different numbers of attribute aspects.

Table 9: Zero-shot lesion-attribute matching performance (%) of **Malenia**.

| Attribute Aspect | MSD | | | | KiTS23 | | In-house Dataset | |
| --- | --- | --- | --- | --- | --- | --- | --- | --- |
| | Hepatic Vessel Tumor | | Pancreas Cyst | | Kidney Tumor | | Liver Cyst | |
| | Precision↑ | Recall↑ | Precision↑ | Recall↑ | Precision↑ | Recall↑ | Precision↑ | Recall↑ |
| Enhancement Status | 100 | 100 | 100 | 100 | 100 | 100 | 100 | 100 |
| Location | 99.34 | 99.67 | 98.46 | 96.92 | 99.68 | 99.25 | 100 | 100 |
| Shape | 95.71 | 97.03 | 89.23 | 90.77 | 94.27 | 93.67 | 86.67 | 90.00 |
| Relationship with Surrounding Organs | 99.01 | 98.35 | 96.92 | 98.45 | 99.18 | 98.57 | 93.34 | 96.67 |
| Density | 92.41 | 93.40 | 92.31 | 93.76 | 94.35 | 94.87 | 92.63 | 92.11 |
| Density Variations | 95.62 | 96.13 | 93.04 | 92.81 | 94.52 | 93.79 | 90.00 | 93.34 |
| Surface Characteristics | 91.05 | 91.62 | 89.83 | 90.45 | 92.44 | 93.83 | 88.21 | 88.06 |
| Specific Features | 84.54 | 85.46 | 82.13 | 82.40 | 83.29 | 82.96 | 82.75 | 83.34 |

# E    EVALUATION OF LESION-ATTRIBUTE MATCHING

Mask-attribute alignment is the core foundation of our method. In this section, we report the zero-shot lesion-attribute matching performance of **Malenia**. Table 9 presents the lesion-level mask-attribute matching precision and recall for each attribute aspect on the zero-shot testing dataset. **Malenia** achieved outstanding zero-shot performance in the task of lesion-attribute alignment. Notably, we observed that for attributes that are relatively easier to determine, such as "Enhancement Status" and "Location", the model attained nearly 100% precision and recall. However, for attributes requiring more complex visual semantic understanding, such as "Surface Characteristics", "Density", and "Specific Features", there is still room for improvement. This may represent a meaningful research direction for future work. In Fig. 6, we present visualizations of some of Malenia's segmentation and attribute alignment results.

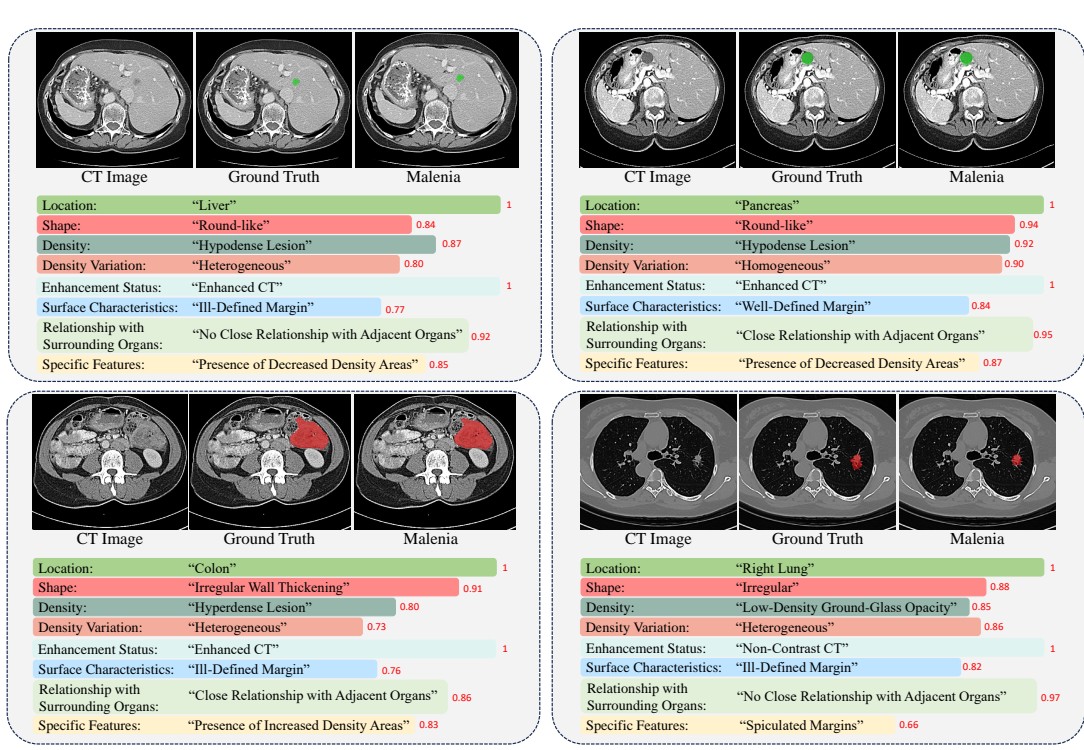

Figure 6: Visualization of segmentation and lesion-attribute matching results of **Malenia**. The numbers indicate the normalized similarity scores between the lesion masks and their corresponding attributes.

Figure 7: Visualization illustrating how utilizing both mask and text embeddings improves segmentation performance for unseen and seen lesions with ambiguous boundaries. We present several prediction cases generated using both text embeddings and mask tokens, text embeddings only (TE), and mask tokens only (MT).

# F   ANALYSIS OF MODEL ROBUSTNESS

## F.1   HANDLING LESIONS WITH AMBIGUOUS BOUNDARIES

Fig. 7 provides detailed examples illustrating how the CMKI module integrates text tokens and mask tokens to enhance the performance for lesions with ambiguous boundaries. We observed that when lesion boundaries are particularly blurry (as seen in the Hepatic Vessel Tumor cases on the left side of Fig. 7), the visual features from the mask tokens fail to accurately capture the lesion's area. In such instances, text descriptions of attributes—such as the lesion's shape and surface characteristics—provide additional information. The segmentation generated by the corresponding text embeddings refines the results produced by mask tokens alone, leading to improved performance.

Additionally, when encountering lesions with highly atypical appearances, such as colon tumors, which exhibit significant variability in shape and size as well as poorly defined boundaries (as seen in the colon tumor cases on the right side of Fig. 7), we observed that the model tends to produce more false positives when relying solely on mask tokens for segmentation predictions. Due to the clear semantic information and specific context provided by distinct attribute aspects, such as density, density variations, and surface texture, segmentation results generated using attribute embeddings exhibit significantly fewer false positives. These examples demonstrate how our framework leverages the strengths of text embeddings and mask tokens to address ambiguous or borderline cases, such as lesions with poorly defined boundaries or highly atypical appearances.

## F.2   HANDLING INCORRECT TEXT INPUTS

During inference, we use all stored attribute embeddings. However, users may wish to specify their own attributes during testing, which could lead to the inclusion of incorrect attributes. In this section, we provide an example to illustrate how our model handles ambiguous or incorrect text inputs in Fig. 8. Specifically, we present two scenarios. (1) The model has access to both correct and incorrect text inputs (Fig. 8a). In this case, **Malenia** continues to produce stable segmentation results. Owing to our proposed mask-attribute alignment strategy, **Malenia** accurately computes the similarity between different text input embeddings and the mask tokens, effectively filtering out incorrect text inputs and selecting the best-matched attributes. (2) The model only has access to incorrect text inputs from users (Fig. 8b). For example, the image may contain a lung tumor in the right lung, but the user provides an attribute for the left lung. In this case, we visualized both the segmentation heatmap generated by the text embeddings and the one generated by the mask tokens. As shown in (Fig. 8b), although the text inputs are incorrect, the similarity between these inputs and the

Table 10: Computational cost comparison between **Malenia** and competing methods used in our evaluation experiments. The FLOPs are computed based on an input with a spatial size of $96 \times 96 \times 96$ on the same A100 GPU.

| Efficiency \ Method | **Malenia** | ZePT | H-SAM | nnU-Net | Swin UNETR |
|---|---|---|---|---|---|
| Params | 323.86M | 745.94M | 557.60M | 370.74M | 371.94M |
| FLOPs | 1128.96G | 1337.59G | 3643.96G | 6742.36G | 2005.48G |

image features remains very low, resulting in segmentation heatmaps with low confidence. In contrast, the mask tokens continue to produce accurate segmentation heatmaps with high confidence. Thus, during the ensembling process, the high-confidence heatmaps from mask tokens prevent Malenia from being influenced by incorrect text inputs.

## G    COMPUTATIONAL COST COMPARISON

We compare the number of parameters across different methods to evaluate computational efficiency. Moreover, we also assessed the inference speed of various models, as this is crucial for clinical applications. We utilized floating-point operations per second (FLOPs) as a metric to measure the inference speed. Table 10 presents the number of parameters and FLOPs (G) for **Malenia** and the comparison models, demonstrating that our method not only achieves the best performance but also superior computational efficiency. Remarkably, due to the mask-attribute alignment mechanism, **Malenia** directly matches mask embeddings with stored attribute embeddings and does not require an additional text encoder to process text inputs during inference. This makes it more flexible and efficient than language-driven segmentation models like ZePT (Jiang et al., 2024), which rely on user-provided text prompts based on the input images. Additionally, **Malenia** does not require an extra encoder to process complex visual prompts. Instead, it directly generates segmentation predictions for the input image while also producing attribute descriptions. This approach is more aligned with real clinical diagnostic scenarios, where doctors expect the AI model to first provide diagnostic predictions to assist in the diagnostic process.

## H    DISCUSSION ON DIFFERENCES BETWEEN ZEPT AND MALENIA

The primary distinctions between our approach and ZePT (Jiang et al., 2024) are as follows:

- **Alignment Strategy**. ZePT Jiang et al. (2024) uses a single-scale mask-text alignment, aligning only the image features from the final Transformer decoder block with a basic textual description. In contrast, our method employs a multi-scale mask-attribute alignment, aligning multi-scale image features with detailed lesion attribute descriptions through a multi-positive contrastive loss. This represents a key technical advancement of our approach. The alignment strategy used by ZePT misses the opportunity to leverage multi-scale features, which hinders its ability to capture targets with significant scale variations. Furthermore, ZePT treats each mask and its corresponding category as the only positive sample pair, resulting in weak associations between different categories within the learned feature space. For instance, in ZePT, the mask corresponding to a liver tumor and the text embeddings of a liver cyst are treated as negative sample pairs, despite the shared attributes between liver tumors and liver cysts. Such similarities are overlooked by ZePT, whereas in our approach, these similarities are explicitly captured through attribute alignment. Therefore, Malenia demonstrates stronger generalization capabilities.
- **Mask Prediction**. ZePT Jiang et al. (2024) relies solely on mask tokens for segmentation mask prediction. In contrast, our approach introduces a Cross-Modal Knowledge Injection (CMKI) module,

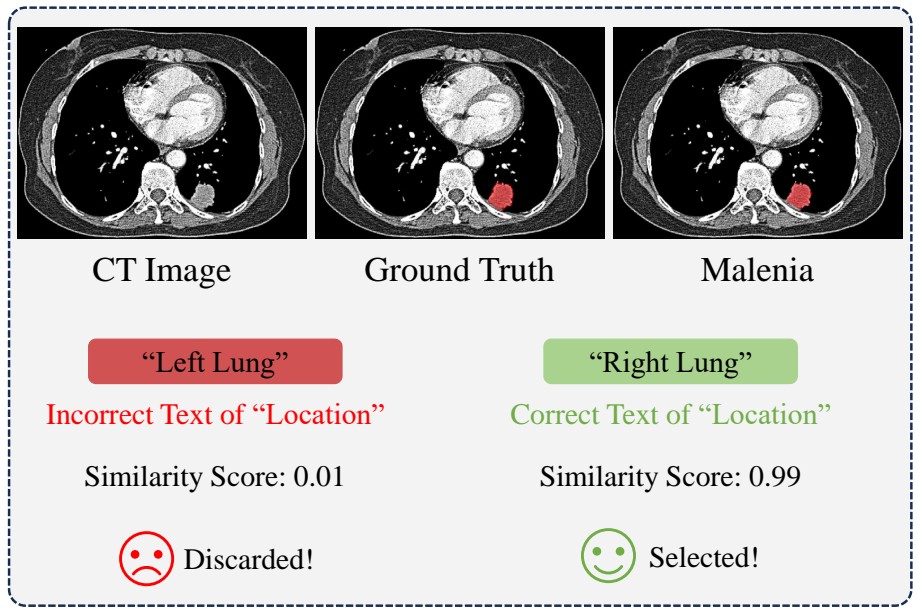

(a) Both correct and incorrect attribute texts are provided

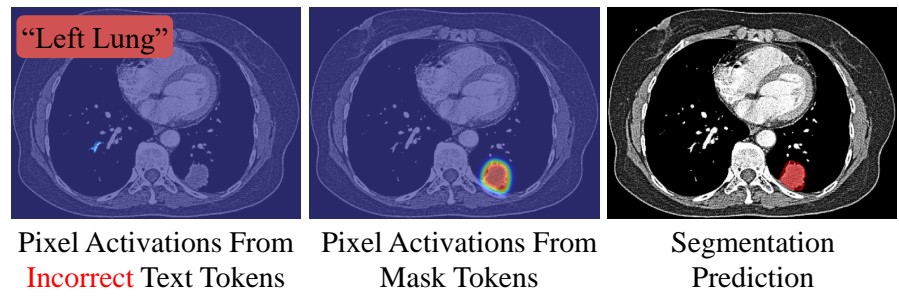

(b) Only incorrect attribute texts are provided

Figure 8: Visualization of how **Malenia** handles incorrect text inputs.

Table 11: Details of the clinical knowledge table used during inference. We present several common diseases along with their corresponding attributes. We will open-source all diseases involved in this study along with their corresponding attributes.

| Lesion Type | Location | Shape | Density | Relationship with Surrounding Organs |
|---|---|---|---|---|
| Hepatic Vessel Tumor | Liver | Round-like, Irregular | Hypodense fluid-like lesion& Isodense lesion& Hyperdense lesion | No close relationship with surrounding organs, Close relationship with adjacent organs |
| Pancreas Cyst | Pancreas | Cystic | Hypodense fluid-like lesion | No close relationship with surrounding organs |
| Kidney Tumor | Kidney | Round-like & Irregular | Hypodense fluid-like lesion & Isodense lesion & Hyperdense lesion | No close relationship with surrounding organs & Close relationship with adjacent organs |
| Liver Cyst | Liver | Cystic | Hypodense fluid-like lesion | No close relationship with surrounding organs |
| Kidney Stone | Kidney | Nodular | Hyperdense lesion | No close relationship with surrounding organs |
| Gallbladder Tumor | Gallbladder | Round-like & Irregular | Hypodense lesion & Hyperdense lesion & Isodense lesion | No close relationship with surrounding organs & Close relationship with adjacent organs |

| Lesion Type | Density Variation | Enhancement Status | Surface Characteristics | Specific Features |
|---|---|---|---|---|
| Hepatic Vessel Tumor | Homogeneous & Heterogeneous | Enhanced CT & Non-contrast CT | Well-defined margin & Ill-defined margin | Presence of decreased density areas & Presence of increased density areas |
| Pancreas Cyst | Homogeneous | Enhanced CT& Non-contrast CT | Well-defined margin | Cyst |
| Kidney Tumor | Homogeneous & Heterogeneous | Enhanced CT & Non-contrast CT | Well-defined margin & Ill-defined margin | Presence of decreased density areas & Presence of increased density areas |
| Liver Cyst | Homogeneous | Enhanced CT & Non-contrast CT | Well-defined margin | Cyst |
| Kidney Stone | Homogeneous | Enhanced CT & Non-contrast CT | Well-defined margin | Stone |
| Gallbladder Tumor | Homogeneous & heterogeneous | Enhanced CT & Non-contrast CT | Well-defined margin & Ill-defined margin | Stone |

which integrates features from both vision and language modalities, utilizing both text tokens and mask tokens for segmentation mask prediction. We demonstrate in Fig. 7 that CMKI is particularly effective in zero-shot lesion segmentation tasks, where unseen categories can be recognized by leveraging the complementary strengths of both vision and language modalities.

Thus, the main technical contributions of our work are: (1) Leveraging multi-scale features for cross-modal alignment. (2) Decomposing textual descriptions/reports into categorized attributes. (3) Introducing a multi-positive alignment mechanism that establishes better cross-modal feature representations. (4) Developing the Cross-Modal Knowledge Injection module. Our ablation studies also demonstrate that incorporating these designs significantly enhances zero-shot generalization performance. Furthermore, Fig. 7 shows that leveraging fine-grained textual information facilitates tumor segmentation, especially in scenarios where visual cues are limited.

# I    LIMITATIONS AND FUTURE WORK

Malenia currently demonstrates superior zero-shot lesion segmentation performance, nearing the performance of fully supervised, task-specific models. However, several limitations persist, revealing valuable directions for future research:

- Cross-domain and cross-modality challenges: Zero-shot lesion segmentation across anatomical regions and imaging modalities remains highly challenging. For instance, training the model on lesions in abdominal CT data and directly testing it on brain cancer MRI is challenging due to significant differences in anatomical structures and feature distributions between the training and testing images. Achieving robust cross-domain and cross-modality zero-shot generalization is a complex but essential research direction in both natural and medical imaging domains.

- Scalability to other imaging modalities: This work focuses primarily on 3D CT scans. While radiology reports for other modalities, such as X-ray and MRI, also contain structured attributes that could be leveraged, Malenia's applicability to these modalities has not yet been tested. Future work will focus on adapting the framework to handle these imaging modalities, addressing their unique characteristics such as spatial resolution and semantic differences.

- Handling lesions with ambiguous boundaries: Defining the boundaries of lesions with unclear or diffuse margins, such as pancreatic cancer or colon cancer, is inherently challenging and subjective even for radiologists. Our analysis in Table 9 indicates that Malenia's mask-attribute matching performance on attributes such as Surface Characteristics still has room for improvement, as the inherent uncertainty in the corresponding ground truth labels affects segmentation accuracy. Future efforts will focus on enhancing the model's ability to handle these ambiguous cases, potentially incorporating additional contextual information.

- Capturing complex visual semantics: Malenia performs less effectively on attributes requiring intricate visual semantic understanding, such as Surface Characteristics and Specific Features. The challenges here may arise from the variability and imbalance in specific feature distributions, as well as the inherent difficulty in categorizing rare or atypical features. Exploring advanced model architectures and integrating additional contextual information will be crucial for enhancing performance on these attributes.

- Generalization to diverse lesion types: While Malenia demonstrates strong performance on 12 lesion categories, the diversity of lesions in clinical practice is vast. Future work will focus on evaluating Malenia on a broader range of lesion types to further validate its generalization capabilities and ensure applicability to diverse clinical scenarios.

We believe addressing these limitations will significantly enhance Malenia's robustness, scalability, and clinical utility, contributing to the broader advancement of zero-shot learning in medical imaging.

