# OpenReview forum: "Unleashing the Potential of Vision-Language Pre-Training for 3D Zero-Shot Lesion Segmentation via Mask-Attribute Alignment"
_ICLR.cc/2025/Conference — ICLR 2025 Poster_

### Official Review · Reviewer_eodD · 2024-10-30

**Soundness:** 2
**Presentation:** 2
**Contribution:** 2
**Rating:** 8
**Confidence:** 4

**Summary:**

The paper introduces Malenia, a novel framework specifically designed for 3D zero-shot lesion segmentation in medical images. The authors aim to address the challenges of transferring image-level knowledge to pixel-level tasks—such as lesion segmentation—by proposing a multi-scale lesion-level mask-attribute alignment approach. Malenia improves the alignment between the visual features of unseen lesions with textual representations, leveraging Cross-Modal Knowledge Injection (CMKI) to enhance both visual and textual embeddings. Experimental results across three datasets demonstrate that Malenia outperforms state-of-the-art (SOTA) methods in zero-shot lesion segmentation.

**Strengths:**

1. The proposed Malenia framework is novel, particularly in its use of multi-scale mask-attribute alignment for 3D lesion segmentation. This is a meaningful extension of zero-shot segmentation methods to the medical domain, where unseen lesion categories are common (e.g., "hepatocellular carcinoma," mentioned in Sec. 1).
2. The CMKI module is a strong contribution because it combines visual and textual embeddings, enriching the segmentation prediction with complementary information from both modalities. This is a novel approach that enhances the zero-shot capability of the model.
3. The paper introduces a multi-positive contrastive loss to fine-tune the alignment between lesion masks and textual attributes. This approach allows the model to generalize better to unseen lesions by learning from shared visual attributes, which distinguishes it from prior works like SAM (Kirillov et al., 2023).

**Weaknesses:**

1. The method relies on transforming patient reports into structured descriptions of eight visual attributes, which requires expert radiologist involvement. This process may not be scalable or practical in real-world applications where such annotations are not readily available.
2. As noted in Appendix D, the model performs less effectively on attributes requiring complex visual semantic understanding, such as "Surface Characteristics" and "Specific Features". This suggests a potential limitation in capturing intricate visual patterns.
3. While the model shows strong performance on 12 lesion categories, the diversity of lesions in clinical practice is vast. The ability of Malenia to generalize to a wider range of unseen lesions without additional training remains uncertain.

**Questions:**

1. In Section 3.1, the authors employ the Multi-Positive InfoNCE (MP-NCE) loss for aligning mask embeddings with multiple attribute embeddings. Could the authors provide more details on how the MP-NCE loss is computed and how it facilitates learning with multiple positive pairs? Specifically, how are the positive and negative pairs defined in the context of multiple attributes?
   Could the authors explain how this formulation effectively handles multiple positive pairs for each mask token?
2. The model uses a fixed number of mask tokens ($N=16$) and defines $R=8$ attribute aspects. Have the authors conducted any experiments to assess how varying $N$ or the number of attributes affects the model's performance? An ablation study on these hyperparameters could provide insights into the model's sensitivity and optimal settings.
3. As noted in Appendix D, the model performs less well on attributes like "Surface Characteristics" and "Specific Features". What are the potential reasons for this lower performance? Could incorporating additional contextual information or advanced features help the model better capture complex visual semantics?

---

> ### Author Response · Authors · 2024-11-24
> **Response to Reviewer eodD (1/3)**
>
> Thank you very much for recognizing the novelty of our framework and the significance of our technical contributions: "The proposed Malenia framework is novel..., a meaningful extension of zero-shot segmentation methods...; The CMKI module is a strong contribution..., a novel approach that enhances the zero-shot capability of the model."
>
> >**Q1**. The method relies on transforming patient reports into structured descriptions of eight visual attributes, which requires expert radiologist involvement. This process may not be scalable or practical in real-world applications where such annotations are not readily available.
>
> We fully agree with your comment and have decided to open-source all the annotated radiology reports and lesion attribute descriptions related to the public datasets used in this work to benefit the research community.
> Training vision-language models or report generation models using these annotations can facilitate attribute predictions on unlabeled data. Human experts can then review and correct these predictions, potentially reducing labor costs, saving time, and ultimately enhancing scalability.
>
> >**Q2**. As noted in Appendix D, the model performs less effectively on attributes requiring complex visual semantic understanding, such as "Surface Characteristics" and "Specific Features". This suggests a potential limitation in capturing intricate visual patterns.
> What are the potential reasons for this lower performance? Could incorporating additional contextual information or advanced features help the model better capture complex visual semantics?
>
> Thank you for pointing out the model's performance limitations on attributes requiring complex visual semantic understanding, such as "Surface Characteristics" and "Specific Features". We appreciate this observation and agree that capturing intricate visual patterns remains a significant challenge.
>
> We believe that the challenges with "Surface Characteristics" may be attributed to the inherent difficulty of recognizing whether lesion boundaries are blurred, a task that is extremely challenging even for radiologists to quantify precisely. For such complex semantic features, the uncertainty of the model's predictions tends to rise.
>
> For "Specific Features," the issue likely stems from the diversity and rarity of these features across different lesions. Many specific features cannot be easily categorized into common attributes, resulting in significant variability and highly imbalanced distributions. Some types of specific features may occur very infrequently, further impacting the model’s performance on this attribute.
>
> We fully agree with your suggestion of incorporating additional contextual information or advanced features as a potential avenue to address these challenges. We plan to explore this direction in future iterations of the model and sincerely thank you for this valuable input. In response, we have incorporated this limitation into Section I: LIMITATIONS AND FUTURE WORK, where we discuss these issues and outline strategies for improving Malenia’s ability to handle complex semantic attributes in future work.
>
> >**Q3**. While the model shows strong performance on 12 lesion categories, the diversity of lesions in clinical practice is vast. The ability of Malenia to generalize to a wider range of unseen lesions without additional training remains uncertain
>
> Thank you for raising the concern regarding the generalizability of Malenia to a broader range of unseen lesions in clinical practice.
> The disease attributes we selected were the result of thorough discussions with radiologists, aiming for attributes that are sufficiently generalizable across a wide range of diseases. Theoretically, Malenia should be able to generalize to diseases with similar attributes to those defined.
> While our model demonstrates strong performance on the 12 lesion categories tested,
> we recognize that the diversity of lesions in real-world clinical scenarios is much broader. To address this, we have added a discussion to Section I: LIMITATIONS AND FUTURE WORK, acknowledging this limitation and outlining plans to evaluate Malenia on a wider variety of lesion types in future work. This will help further validate its generalization capabilities and ensure its applicability to diverse clinical cases.

---

> ### Author Response · Authors · 2024-11-24
> **Response to Reviewer eodD (2/3)**
>
> >**Q4**. In Section 3.1, the authors employ the Multi-Positive InfoNCE (MP-NCE) loss for aligning mask embeddings with multiple attribute embeddings. Could the authors provide more details on how the MP-NCE loss is computed and how it facilitates learning with multiple positive pairs? Specifically, how are the positive and negative pairs defined in the context of multiple attributes? Could the authors explain how this formulation effectively handles multiple positive pairs for each mask token?
>
> Thank you for your interest in the technical details of our method.
> Specifically, for the $i$-th mask token $m_i$, its positive pairs are formed by itself and
> the corresponding attributes from each of the eight attribute categories. Each attribute category
> contains multiple distinct attributes, and one attribute from
> each attribute category forms a positive sample pair with $m_i$, resulting in a total of eight positive sample pairs.
> All other attributes that are not related to $m_i$ constitute negative sample pairs.
>
> In traditional vision-language models, such as CLIP and ZePT, visual features typically form a positive sample pair only with the text embeddings of their corresponding category, while forming negative sample pairs with text embeddings from other categories.
> As a result, these models use InfoNCE loss, which involves a single positive pair and multiple negative pairs.
> Let the similarity score between the $i$-th mask token and $j$-th text embedding
> be denoted as $S_{i,j} > 0$. Let $P_i$ be the set of all positive sample
> indices of the $i$-th mask token and $N_i$ be the set of all negative sample indices of the
> i-th mask token.
> For example, if there is only one positive sample for each sample in a batch,
> say $P_i = \\{p_i\\}$.
> Then the InfoNCE loss for the $i$-th mask token
> can be described by:
> $$
>   \\mathcal L_i = - \\log \\frac{S_{i,p_i}}{S_{i,p_i} + \\sum_{n \\in \\mathbb N_i} S_{i,n}}
> $$
>
> In our method, since each mask token has multiple positive pairs, we employ the Multi-Positive InfoNCE (MP-NCE) loss.
> the MP-NCE loss for the $i$-th mask token
> can be described by:
>
> $$
>   \\mathcal L_i = \\mathbb E_{p \\in P_i}  \\Big[ - \\log \\frac{S_{i,p}}{S_{i,p} + \\sum_{n \\in \\mathbb N_i} S_{i,n}}\\Big]
> $$
>
> MP-NCE loss incorporates multiple positive pairs by averaging their individual
> losses, optimizing the similarity between each mask token and its corresponding positive pairs, thereby facilitating learning with multiple positive pairs.
>
> We hope that the simplified loss formula for each mask token provided above helps you understand how MP-NCE loss effectively handles multiple positive pairs for each mask token. If you have further questions, we would be happy to discuss them with you.

---

> ### Author Response · Authors · 2024-11-24
> **Response to Reviewer eodD (3/3)**
>
> >**Q5**. The model uses a fixed number of mask tokens ($N = 16$) and
> defines $R = 8$ attribute aspects.
> Have the authors conducted any experiments to assess how varying
>  or the number of attributes affects the model's performance? An ablation study on these hyperparameters could provide insights into the model's sensitivity and optimal settings.
>
> Thank you for your valuable and constructive suggestions, which have helped further strengthen the experimental section of our paper. We have conducted ablation studies on the number of mask tokens and the number of attributes, and these results have been added to Section D.2 and D.3 in Appendix of the revised version.
>
> Regarding the number of mask tokens, we found that when $N$ is less than 12, the model's performance drops sharply. This observation is consistent with previous conclusions from MaskFormer-based methods [1], which suggest that the number of mask tokens should ideally be greater than the number of possible or useful classes in the data. When $N$ exceeds 16, we observed no significant performance improvement. Therefore, we set the default value to $N=16$.
>
> Regarding the number of attribute categories, we observed that as we incrementally increased the number of attribute categories from 1 to 8, the model's performance consistently improved with greater attribute diversity.
> This result demonstrates the superiority of our approach in using multiple attribute aspects to describe lesions from different perspectives, providing valuable textual features that help capture subtle visual characteristics of the lesions.
> This finding also supports your earlier suggestion that incorporating additional contextual information or advanced features may enhance the model's ability to capture complex visual semantics.
> Given that defining and annotating more attribute categories requires additional expertise and incurs higher annotation costs, we will strive to expand the range of attributes and explore solutions to address the challenges of annotation costs and scalability in the future.
>
> **Reference**
> 1. Cheng, Bowen, et al. "Masked-attention mask transformer for universal image segmentation." CVPR, 2022.

---

> ### Comment · Reviewer_eodD · 2024-11-24
>
> Zero-shot segmentation of medical images is a highly challenging task, and aligning text and image features beyond the instance level has been a persistent focus of research. Considering that the authors plan to open-source the annotated dataset presented in this paper and have successfully addressed the concerns I raised, I believe this work, after revision, could have a meaningful impact on the community. As such, I am willing to raise my score to 8 in support of this research.
>
> That said, I have a few minor suggestions. The fonts and color schemes used in the main figures (Fig. 1 and Fig. 2) are somewhat small and appear a bit thin, which made it slightly difficult for me to read, especially when I printed the paper for review. For example, some key components, such as the **Multi-Positive Contrastive module**, could be emphasized more. To improve the overall clarity and impact of the paper, I recommend refining the structure of these diagrams, perhaps by redrawing the block diagrams with clearer fonts and better visual emphasis on the main modules.

---

> > ### Author Response · Authors · 2024-11-26
> > **Response to Reviewer eodD**
> >
> > We sincerely appreciate your recognition of the value and contributions of our work. In the further revised version of the paper, we have adjusted the font sizes in Figures 1 and 2 and highlighted our key proposed modules. We are grateful for all your valuable suggestions, which have significantly improved both the completeness of our experiments and the overall presentation of the paper. We will continue our efforts and hope to contribute meaningfully to the research community.

---

### Official Review · Reviewer_DS2M · 2024-11-01

**Soundness:** 3
**Presentation:** 3
**Contribution:** 3
**Rating:** 6
**Confidence:** 5

**Summary:**

The paper introduces "Malenia," a framework for zero-shot 3D lesion segmentation using vision-language pre-training methods. This framework aligns multi-scale mask representations of lesions with textual embeddings of disease attributes, which facilitates the segmentation of unseen lesion types in medical imaging.

**Strengths:**

1. This study bridges the gap between mask representations and textual disease attributes for zero-shot lesion segmentation.
2. The empirical validation includes comprehensive experiments across multiple datasets and lesion types, demonstrating the robustness and effectiveness of Malenia.

**Weaknesses:**

1. While the paper effectively addresses 3D CT scans, it does not discuss the applicability of the Malenia framework to other imaging modalities like X-ray.
2. The paper does not discuss how the framework handles ambiguous or borderline cases where lesion boundaries are not well-defined or where lesions exhibit highly atypical appearances.
3. There is no thorough discussion on the limitations or failures of the proposed model.

**Questions:**

1. Could the authors provide insights on the computational efficiency of their framework?
2. This work and its motivation are similar to the previous works [1-2], please describe the differences between them.
[1] Li, Z., Li, Y., Li, Q., Wang, P., Guo, D., Lu, L., ... & Hong, Q. (2023). Lvit: language meets vision transformer in medical image segmentation. IEEE transactions on medical imaging.
[2] Huang, X., Li, H., Cao, M., Chen, L., You, C., & An, D. (2024). Cross-Modal Conditioned Reconstruction for Language-guided Medical Image Segmentation. arXiv preprint arXiv:2404.02845.

---

> ### Author Response · Authors · 2024-11-24
> **Response to Reviewer DS2M (1/3)**
>
> We appreciate your thoughtful feedback and accolades on our contribution :"This study bridges the gap between...; ...comprehensive experiments...demonstrating the robustness and effectiveness of Malenia".
>
> >**Q1**. While the paper effectively addresses 3D CT scans, it does not discuss the applicability of the Malenia framework to other imaging modalities like X-ray.
>
> Thank you for highlighting the limitation regarding the applicability of the Malenia framework to other imaging modalities.
> In this work, we focus primarily on 3D CT scans. We recognize the importance of extending Malenia to handle other modalities such as X-ray and MRI.
> Given that radiology reports for these modalities can also be structured into meaningful attributes, we anticipate that Malenia could demonstrate effective performance in these domains as well.
> We have added a discussion of this limitation and outlined future research directions in the LIMITATIONS AND FUTURE WORK section I of the revised manuscript.
>
> >**Q2**. The paper does not discuss how the framework handles ambiguous or borderline cases where lesion boundaries are not well-defined or where lesions exhibit highly atypical appearances.
>
> Thank you for highlighting the important concern regarding our framework’s performance in handling lesions with unclear boundaries or highly atypical appearances.
> We acknowledge that defining lesion boundaries is a critical and challenging task.
>
> We provide visualization examples in Fig. 6 in the Appendix of the revised paper to illustrate how the framework handles lesions with ambiguous boundaries or highly atypical appearances. This is primarily achieved through the proposed cross-modal knowledge injection module, which combines visual and language features.
>
> We observed that when lesion boundaries are particularly blurred, the visual features from mask tokens fail to accurately capture the lesion area. In such cases, text descriptions of attributes—such as the lesion’s location, shape, and surface characteristics—provide additional information to aid in tumor segmentation. Additionally, when dealing with lesions that have highly atypical appearances, such as colon tumors with significant variability in shape and size and poorly defined boundaries, the model tends to produce more false positives when relying solely on mask tokens for segmentation predictions. In contrast, incorporating the clear semantic information and specific context provided by distinct attribute aspects, such as density, density variations, and surface texture, results in significantly fewer false positives in the segmentation.
>
> However, we acknowledge that this approach cannot address all borderline cases. We have expanded the discussion in Section I in Appendix to include the limitations of Malenia in handling lesions with ambiguous boundaries.
> We fully agree that this is a meaningful direction for future research and hope our response addresses your concerns.

---

> ### Author Response · Authors · 2024-11-24
> **Response to Reviewer DS2M (2/3)**
>
> >**Q3**. There is no thorough discussion on the limitations or failures of the proposed model.
>
> Thank you for pointing out the need for a more thorough discussion of the limitations and potential failures of our proposed model. In response, we have expanded the discussion in Section I: LIMITATIONS AND FUTURE WORK,
> providing a more comprehensive analysis of the current shortcomings, including:
> - **Challenges in Achieving Cross-Modality Zero-Shot Lesion Segmentation.**
> - **Current Limitations of Model Development Focused Solely on CT Modalities.**
> - **Potential Performance Issues with Segmenting Lesions with Ambiguous Boundaries.**
>
> We hope this additional discussion addresses your concern and provides a clearer understanding of the limitations and future directions for this work.
>
> >**Q4**. Could the authors provide insights on the computational efficiency of their framework?
>
> Thank you for your interest in the computational efficiency of our framework. A detailed comparison of the computational cost between our proposed model and other methods is provided in Table 10 of the revised paper, which highlights the efficiency of our approach relative to existing frameworks. Compared to existing vision-language models, such as ZePT, our method directly matches mask embeddings with stored attribute embeddings and does not require an additional text encoder to process text inputs during inference. Furthermore, our method does not require an extra vision encoder to handle complex visual prompts, resulting in fewer model parameters and reduced GFLOPs. We hope this addresses your concern.

---

> ### Author Response · Authors · 2024-11-24
> **Response to Reviewer DS2M (3/3)**
>
> >**Q5**. This work and its motivation are similar to the previous works [1-2], please describe the differences between them. [1] Li, Z., Li, Y., Li, Q., Wang, P., Guo, D., Lu, L., ... & Hong, Q. (2023). Lvit: language meets vision transformer in medical image segmentation. IEEE transactions on medical imaging. [2] Huang, X., Li, H., Cao, M., Chen, L., You, C., & An, D. (2024). Cross-Modal Conditioned Reconstruction for Language-guided Medical Image Segmentation. arXiv preprint arXiv:2404.02845.
>
> Thank you for sharing Lvit and RecLMIS with us. Their relevance to our work is significant and appreciated.
> We have now included a discussion of
> these two works in Section 2 of the revised paper. These works share a level of similarity with ours,
> aiming to develop medical AI models that incorporate text information
> and explore cross-modal interactions to segment multiple anatomical structures.
> The differences between these methods and ours can be summarized in two main aspects:
>
> 1. These methods are 2D segmentation models focusing on CT slices of specific body parts, such as the chest, whereas our approach is a 3D segmentation model aimed at segmenting lesions across various anatomical regions.
>
> 2. These two methods mainly focus on fully-supervised and semi-supervised settings, emphasizing the enhancement of segmentation performance for seen categories by combining visual and language features.
> In contrast, our work addresses the zero-shot setting, aiming to develop generalized and robust textual features, such as disease attributes, along with a novel vision-language alignment strategy to
> enhance segmentation performance for unseen lesions.

---

> ### Comment · Reviewer_DS2M · 2024-11-26
>
> Thanks for the revision. The authors have addressed most of my concerns. However, the author should also discuss how the framework handles ambiguous or incorrect text input. It is crucial for the model's robustness. I will maintain my original rating and look forward to further revisions.

---

> > ### Author Response · Authors · 2024-11-26
> > **Response to Reviewer DS2M**
> >
> > Thank you for these valuable suggestions! In the further revised version, we have added Section F.2 and Figure 8 to illustrate how our model handles incorrect text inputs. We specifically discuss two scenarios: the first is when the model has access to both correct and incorrect text inputs, and the second is a more challenging scenario where the model only has access to incorrect text inputs.
> >
> > In the first scenario, our method accurately computes the similarity between different text input embeddings and mask tokens, effectively filtering out incorrect inputs and selecting the best-matched attributes. In the second scenario, we visualize the segmentation heatmaps to show how the mask tokens overcome the errors introduced by incorrect text inputs. We appreciate your suggestions, as these experiments further demonstrate the robustness of our method.

---

### Official Review · Reviewer_thtg · 2024-11-03

**Soundness:** 2
**Presentation:** 4
**Contribution:** 2
**Rating:** 5
**Confidence:** 5

**Summary:**

This paper introduces Malenia, a framework for zero-shot lesion segmentation in 3D medical images, aimed at addressing the challenge of transferring image-level knowledge to pixel-level segmentation tasks. Building on advancements in medical vision-language pre-training, Malenia integrates a multi-scale mask-attribute alignment framework and a Cross-Modal Knowledge Injection module to link visual and textual features, enabling the model to handle previously unseen lesions. The authors evaluate Malenia's performance through experiments on three datasets (two public and one private) across 12 lesion categories.

**Strengths:**

1. The topic on zero-shot lesion segmentation is valuable, as clinical settings often involve diverse, emerging anomalies, and data collection is challenging, increasing the need for models that handle unseen diseases in an open-set context.
2. The authors strengthen their claims with both qualitative and quantitative results, enhancing the credibility of their findings and providing a well-rounded evaluation of the proposed approach.
3. The paper is well-organized, with a clear and logical structure that makes complex concepts accessible and easy to follow for readers.

**Weaknesses:**

1. The Zero-Shot Inference section lacks sufficient detail on how the model handles test CT images containing unseen tumor types. For instance, if a test image includes both a kidney tumor and a gallbladder tumor, it is unclear how the model predicts these different tumors within the same image. Additionally, the phrase “we obtain the class information for each predicted lesion mask by referencing a clinical knowledge table” is ambiguous. Further clarification is needed on what this clinical knowledge table entails and how it is used in the inference process.
2. The comparisons in Table 1 and Table 2 may not be entirely fair. The authors introduce additional ATTRIBUTE DESCRIPTIONS annotations, which provide extra information not available to baseline methods like TransUNet, nnUNet, and Swin UNETR. These baselines rely solely on image data, while the proposed approach leverages attribute annotations, giving it an advantage in performance comparisons.
3. The fully-supervised performance of baseline methods reported in Table 2 appears unusually low. For instance, in the official nnUNet paper, liver tumor segmentation achieved a Dice score of 76 on the test set, whereas the presented result here is only 61.33. Similarly, lung tumor segmentation originally reported a Dice score of 74, but this paper reports 54.44. This discrepancy raises concerns about the reproducibility and fairness of the comparisons.
4. In Table 7, the authors compare their method to existing vision-language pretraining strategies, but it is unclear how these methods were reproduced. Most vision-language pretraining approaches require access to diagnostic reports, which are not included in the public MSD and KiTS23 datasets. It would be helpful to understand how the authors conducted these comparisons without complete diagnostic reports. Additionally, the absence of comparisons with important pretraining models like CT-CLIP weakens the evaluation.
5. Scalability of the proposed method is questionable due to its reliance on ATTRIBUTE DESCRIPTIONS. For practical clinical applications, this approach would require physicians to provide additional lesion descriptions across eight visual attribute aspects, which may not be feasible in all settings. This dependency could limit the method’s applicability in diverse clinical environments.

**Questions:**

please refer to Weaknesses

**Details Of Ethics Concerns:**

This paper utilizes private clinical data, which may require an ethics review to ensure compliance with privacy, security, and safety standards.

---

> ### Author Response · Authors · 2024-11-24
> **Response to Reviewer thtg (1/3)**
>
> Thank you for recognizing the value of our research topic, the quality of our presentation, and the extensiveness of our experiments.
>
> >**Q1**. The Zero-Shot Inference section lacks sufficient detail on how the model handles test CT images containing unseen tumor types. For instance, if a test image includes both a kidney tumor and a gallbladder tumor, it is unclear how the model predicts these different tumors within the same image. Additionally, the phrase “we obtain the class information for each predicted lesion mask by referencing a clinical knowledge table” is ambiguous. Further clarification is needed on what this clinical knowledge table entails and how it is used in the inference process.
>
> We completely agree with your comments and have now revised the related contents. In Section 3.3, we have added a detailed flowchart (Figure 2) to illustrate how our model performs inference when multiple lesion categories are present in an image. Additionally, the details of the Clinical Knowledge Table are provided in Table 11 of Appendix.
>
> To facilitate understanding, we have divided the inference process into four steps for explanation.
> - **Step-I: Mask tokens attend to different regions among the images, partitioning the image into different regions. In this process, different lesion regions in the image are captured by specific mask tokens among the N mask tokens.**
> - **Step-II: We compute the similarity between the N mask tokens and the stored attribute embeddings, after which the mask tokens are matched to the text embeddings based on these similarity scores. Mask tokens representing foreground lesions are matched with the corresponding lesion attributes, while those representing the background are matched with the background text embeddings.**
> - **Step-III: The matched mask tokens and text tokens are fed into the CMKI module, where they are fused through cross-attention. Both are then used to generate mask predictions using scaled dot-product attention with image features. Finally, the mask predictions derived from the mask tokens and text embeddings are combined to produce the final mask prediction.**
> - **Step-IV: The text embeddings for each mask prediction are used to identify the specific category. This is accomplished by querying a Clinical Knowledge Table that records the relationships between each disease and its attributes. Notably, the eight disease attributes we designed are sufficient to uniquely identify the corresponding disease.**
>
> Please kindly refer to Figure 2 in our revised version for a more detailed understanding of these four steps.

---

> ### Author Response · Authors · 2024-11-24
> **Response to Reviewer thtg (2/3)**
>
> >**Q2**. The comparisons in Table 1 and Table 2 may not be entirely fair. The authors introduce additional ATTRIBUTE DESCRIPTIONS annotations, which provide extra information not available to baseline methods like TransUNet, nnUNet, and Swin UNETR. These baselines rely solely on image data, while the proposed approach leverages attribute annotations, giving it an advantage in performance comparisons.
>
> Thank you for your valuable feedback. We would like to clarify that in the ablation study presented in Table 3, we compared the results of training using only ATTRIBUTE DESCRIPTIONS annotations ($S_1$) with those of training using our proposed multi-scale mask-attribute alignment strategy. The results indicate that our strategy significantly outperforms previous approaches, such as those used by ZePT or the Universal Model, which treat ATTRIBUTE DESCRIPTIONS as a single text paragraph to construct only one positive pair. This demonstrates that the performance gains are not solely due to the use of attribute descriptions, but rather due to our strategic approach in effectively leveraging these descriptions.
>
> Regarding the comparison with traditional segmentation models, such as TransUNet, nnUNet, and Swin UNETR, which cannot utilize attribute descriptions, we believe that the ability to incorporate textual features and fuse multimodal information for segmentation inherently represents a significant technical advantage. This perspective has also been acknowledged by reviewers DS2M and eodD as one of the key strengths of our approach. The integration of visual and textual information for joint reasoning is a growing trend, as demonstrated by the Universal model, which leverages the Swin UNETR encoder along with textual information to improve performance.
>
> Similarly, SAM-based models can utilize prompts, whereas traditional methods like TransUNet, nnUNet, and Swin UNETR cannot. Vision-language models, by leveraging additional textual information, inherently contribute to technical advancements. We hope this addresses your concern and clarifies our approach.
>
> >**Q3**. The fully-supervised performance of baseline methods reported in Table 2 appears unusually low. For instance, in the official nnUNet paper, liver tumor segmentation achieved a Dice score of 76 on the test set, whereas the presented result here is only 61.33. Similarly, lung tumor segmentation originally reported a Dice score of 74, but this paper reports 54.44. This discrepancy raises concerns about the reproducibility and fairness of the comparisons.
>
> Thank you so much for your diligent efforts in reviewing our experimental statistics, which have helped us rigorously reassess our experimental setup and evaluation results.
> Regarding the results of MSD liver tumor segmentation, we would like to clarify that the Dice score of 76 in the official nnUNet paper
> was obtained on the test set, while the 61.33 in our paper was obtained through five fold cross-validation on the training cases.
> This is consistent with the results reported in the supplementary materials of the official nnUNet paper [(nnUNet. Nat Methods 2021)](https://www.nature.com/articles/s41592-020-01008-z).
>
> For the MSD lung tumor segmentation results, the Dice score of 74 was obtained on the test set, while the Dice score for five-fold cross-validation on the training cases was reported as 69 in the official nnUNet paper, which is significantly higher than the reproduced score of 54.44 reported in our study. This discrepancy prompted us to reexamine our experiments.
> Upon investigation, we found that during model evaluation, an incorrect parameter setting for clipping intensity during data preprocessing led to generally lower model performance for this task. Specifically, we mistakenly applied the intensity clipping range used for abdominal CTs to lung CTs during inference, despite the significant difference in intensity values between the two types of scans.
> We are grateful for your valuable suggestion, which helped us identify this issue. We sincerely apologize for our previous oversight, and we have corrected the results in Table 2.
>
> | Methods          | MSD-Lung (DSC↑)          | MSD-Lung (NSD↑)          |
> |------------------|--------------------------|--------------------------|
> | TransUNet*       | 67.13 ± 6.08              | 68.89 ± 7.22              |
> | nnUNet*          | 69.50 ± 5.61              | 71.39 ± 6.55              |
> | Swin UNETR*      | 68.95 ± 5.67              | 71.03 ± 6.82              |
> | Universal Model* | 67.27 ± 5.71              | 69.33 ± 6.95              |
> | **Malenia**      | **70.96 ± 5.56**          | **72.34 ± 6.29**          |

---

> ### Author Response · Authors · 2024-11-24
> **Response to Reviewer thtg (3/3)**
>
> >**Q4**. In Table 7, the authors compare their method to existing vision-language pretraining strategies, but it is unclear how these methods were reproduced. Most vision-language pretraining approaches require access to diagnostic reports, which are not included in the public MSD and KiTS23 datasets. It would be helpful to understand how the authors conducted these comparisons without complete diagnostic reports. Additionally, the absence of comparisons with important pretraining models like CT-CLIP weakens the evaluation.
>
> Thank you for your constructive feedback. We fully agree with your suggestion and have included the comparison with CT-CLIP in Table 7 of the revised paper.
> Additionally, we would like to clarify that all diagnostic reports for the public MSD and KiTS23 datasets were annotated by two senior radiologists, as described in Appendix B.2. Consequently, all the vision-language pretraining strategies compared in Table 7 were re-implemented using the official code and pre-trained on the same training data along with the annotated diagnostic reports.
>
> For your convenience, we have also summarized the results of CT-CLIP in the table below.
>
> | Method          | MSD - Hepatic Vessel Tumor (DSC↑) | MSD - Hepatic Vessel Tumor (NSD↑) | MSD - Pancreas Cyst (DSC↑) | MSD - Pancreas Cyst (NSD↑) | KiTS23 - Kidney Tumor (DSC↑) | KiTS23 - Kidney Tumor (NSD↑) | In-house - Liver Cyst (DSC↑) | In-house - Liver Cyst (NSD↑) | In-house - Kidney Stone (DSC↑) | In-house - Kidney Stone (NSD↑) | In-house - Gallbladder Tumor (DSC↑) | In-house - Gallbladder Tumor (NSD↑) |
> |-----------------|----------------------------------|----------------------------------|---------------------------|---------------------------|------------------------------|------------------------------|-----------------------------|-----------------------------|--------------------------------|--------------------------------|-------------------------------------|-------------------------------------|
> | CT-CLIP         | 65.66                            | 75.73                            | 65.85                     | 77.92                     | 74.12                        | 80.24                        | 69.57                       | 75.82                       | 52.45                          | 62.37                          | 58.80                               | 66.84                               |
> | **Malenia**     | **71.96**                         | **81.83**                         | **69.95**                 | **82.00**                 | **76.88**                    | **82.55**                    | **72.44**                   | **79.84**                   | **53.35**                       | **63.69**                       | **64.21**                            | **72.63**                            |
>
>
> >**Q5**. Scalability of the proposed method is questionable due to its reliance on ATTRIBUTE DESCRIPTIONS. For practical clinical applications, this approach would require physicians to provide additional lesion descriptions across eight visual attribute aspects, which may not be feasible in all settings. This dependency could limit the method’s applicability in diverse clinical environments.
>
> We appreciate your thoughtful feedback regarding the scalability of our method, which has prompted us to consider how to better support reproducibility and further development.
> While most CT images in hospitals or other medical data storage centers are typically paired with radiology reports, obtaining both radiology reports and lesion attributes can indeed be challenging and resource-intensive. Therefore, to further support research in this field, we have decided to open-source all the annotated radiology reports and lesion attribute descriptions related to the public datasets used in this work.
> We believe that using our annotations for training vision-language models or report generation models has the potential to reduce the cost of manual annotation and improve the scalability of medical vision-language models in the future.

---

> ### Author Response · Authors · 2024-12-03
> **Any Last-minute Feedback and Thanks to Reviewer thtg**
>
> Dear Reviewer thtg,
>
> As the discussion period nears its deadline, we wanted to reach out for any last-minute feedback you might have. We are very grateful for your important suggestions and diligent efforts in reviewing our paper. We have made every effort to address your concerns in our revised manuscript, including new figures, tables, and revised text to provide a more accurate, comprehensible, and concise presentation within the page limit. We hope that these revisions address your concerns, and we sincerely appreciate all your valuable suggestions.
>
> Authors

---

### Official Review · Reviewer_Lt5s · 2024-11-03

**Soundness:** 3
**Presentation:** 3
**Contribution:** 3
**Rating:** 6
**Confidence:** 3

**Summary:**

This work presents a zero-shot medical image semantic segmentation framework that is based on maskformer and attribute-based visual-language feature fusion. To achieve fine-grain alignment between visual and textual concepts, the authors propose to decompose radiological reports into attributes describing lesions of interest, assisted by LLMs and human annotators. The obtained attributes allows fine-grain fusion and alignment between lesion features and textual concepts, yielding improved generalization performance upon unseen lesions. The proposed framework is evaluated on tumor segmentation tasks, and demonstrates improved segmentation results compared with existing zero-shot segmentation approaches on unseen lesions.

**Strengths:**

The idea of decomposing free-text reports into attributes for fine-grain training and inference is intuitive and feasible. It can significantly reduce the noise and ambiguity associated with unstructured free-text reports and yields improved performance on unseen lesions.

The key components and the detailed implementations are described in good detail.

The proposed approach demonstrates performance improvements compared with previous zero-shot image segmentation methods.

Detailed ablation studies are presented to verify the effectiveness of key components.

**Weaknesses:**

Despite stronger segmentation performance, the major framework looks similar to ZePT (Jiang et al., 2024): maskformer backbone + interacting visual and textual features that are obtained from detailed description of the lesions. Therefore, the key take-home message for readers may be a bit unclear: should the readers interpret the key technical contribution as explicitly decomposing textual descriptions/reports into categorized attributes for the ease of learning and inference?

Readers may argue that using GPT-4 for attribute construction from radiological reports is often infeasible in practice due to the privacy and legal concerns. The authors are encouraged to argue if switching to local open-source LLMs would yield a similar level of performance gain.

The authors may want to decompose some of technical details in Figure 1 into separate figures and move them closer to corresponding paragraphs.

**Questions:**

Considering the practicality issue, would replacing GPT-4 with an open-source LLM for attribute construction (probably also without human annotators?) reaching a similar level of performance gain (as mentioned above)?

---

> ### Author Response · Authors · 2024-11-24
> **Response to Reviewer Lt5s (1/3)**
>
> Thank you so much for recognizing the value of our idea, the completeness of our experiments, and the superiority of our method's performance.
>
> >**Q1**. Despite stronger segmentation performance, the major framework looks similar to ZePT (Jiang et al., 2024): maskformer backbone + interacting visual and textual features that are obtained from detailed description of the lesions. Therefore, the key take-home message for readers may be a bit unclear: should the readers interpret the key technical contribution as explicitly decomposing textual descriptions/reports into categorized attributes for the ease of learning and inference?
>
> Thank you for your valuable feedback regarding our key technical contribution.
> The primary distinctions between our approach and ZePT are as follows:
>
> - **Alignment Strategy**.
> ZePT uses a single-scale mask-text alignment, aligning only the image features from the final Transformer decoder block with a basic textual description. In contrast, our method employs a multi-scale mask-attribute alignment, aligning multi-scale image features with detailed lesion attribute descriptions. This represents a key technical advancement of our approach.
>
> - **Mask Prediction**.
> ZePT relies solely on mask tokens for segmentation mask prediction. In contrast, our approach introduces a Cross-Modal Knowledge Injection (CMKI) module, which integrates features from both vision and language modalities, utilizing both text tokens and mask tokens for segmentation mask prediction.
>
> Thus, the main technical contributions of our work are:
>
> - **Leveraging multi-scale features for cross-modal alignment**.
> - **Decomposing textual descriptions/reports into categorized attributes**.
> - **Introducing a multi-positive alignment mechanism that establishes better cross-modal feature representations**.
> - **Developing the Cross-Modal Knowledge Injection module**.
>
> Our ablation studies demonstrate that incorporating these designs significantly enhances zero-shot generalization performance. Furthermore, Figure 6 of our paper shows that leveraging fine-grained textual information facilitates tumor segmentation, especially in scenarios where visual cues are limited.
>
> We appreciate you pointing out this issue. We agree that in the previous version, these aspects were not sufficiently clear, which may have made it difficult for readers to fully grasp our technical contributions and the distinctions from ZePT.
> Therefore, we have added a discussion section in the Appendix H to further elaborate on these points.

---

> > ### Comment · Reviewer_Lt5s · 2024-11-24
> >
> > I have read through the authors' response and I would like to thank you for the clarification. I agree that compared with ZePT the proposed work leverages multi-scale alignment (essentially deep supervision) and *deep fusion* with attributes in decoding. With that being said, I am still not fully convinced if these improvements would comprise substantial distinctions compared with ZePT, given that these two approaches are commonly adopted for image segmentation / multi-modal fusion.

---

> > > ### Author Response · Authors · 2024-11-25
> > > **Further response to Reviewer Lt5s**
> > >
> > > Thank you so much for your constructive suggestions and diligent efforts in reviewing our revised manuscript! We fully understand and agree with your expectations for a more advanced technical improvement of our work. Your suggestions have prompted us to think more deeply, and we are committed to working even harder in the future.

---

> ### Author Response · Authors · 2024-11-24
> **Response to Reviewer Lt5s (2/3)**
>
> >**Q2**. Readers may argue that using GPT-4 for attribute construction from radiological reports is often infeasible in practice due to the privacy and legal concerns. The authors are encouraged to argue if switching to local open-source LLMs would yield a similar level of performance gain.
> Considering the practicality issue, would replacing GPT-4 with an open-source LLM for attribute construction (probably also without human annotators?) reaching a similar level of performance gain?
>
> Thank you for this valuable suggestion. We completely agree with your comments, particularly regarding the importance of protecting medical data privacy. Based on your suggestion, we conducted additional comparative experiments using MMed-Llama-3-8B [(Qiu et al., Nature Communication 2024)](https://www.nature.com/articles/s41467-024-52417-z), an open-source medical large language model, to extract descriptions of attributes from radiological reports. The results of these experiments have been incorporated into Appendix D.1 Table 8 in the revised version of our paper and are presented in the table below.
>
> | Method                          | Liver Cyst (DSC↑) | Liver Cyst (NSD↑) | Kidney Stone (DSC↑) | Kidney Stone (NSD↑) | Gallbladder Tumor (DSC↑) | Gallbladder Tumor (NSD↑) |
> |---------------------------------|-------------------|-------------------|---------------------|---------------------|--------------------------|--------------------------|
> | MMed-Llama-3-8B                 | 54.97             | 63.06             | 38.66               | 47.71               | 43.28                    | 51.45                    |
> | GPT4                            | 54.98             | 63.08             | 38.67               | 47.73               | 43.28                    | 51.46                    |
> | MMed-Llama-3-8B + Human Annotators | 61.84             | **70.93**         | 43.03               | 52.94               | 47.34                    | **55.79**                |
> | GPT4 + Human Annotators         | **61.85**         | **70.93**         | **43.05**           | **52.95**           | **47.35**                | **55.79**                |
>
>
> The results indicate that using GPT-4 or MMed-Llama-3-8B for attribute extraction from radiological reports, with or without human annotation, has a negligible effect on model performance. This is largely because extracting and structuring attributes already present in the reports is a straightforward and easy task. However, incorporating human annotators significantly enhances performance, as some attributes are not explicitly mentioned in the reports and require expert supplementation—a task beyond the capabilities of current large language models.
>
> Given the high cost of human annotation, and to support future research, we will open-source our attribute annotations for all public datasets used in this study, along with our code, to benefit the research community.
>
> **Reference**
>
> Qiu P, Wu C, Zhang X, et al. Towards building multilingual language model for medicine[J]. Nature Communications, 2024, 15(1): 8384.

---

> > ### Comment · Reviewer_Lt5s · 2024-11-24
> >
> > I have read through the supplemented quantitative results and I would like to thank the authors for the efforts (it appears that human annotations eventually make more observable differences compared to the choice of language models).

---

> ### Author Response · Authors · 2024-11-24
> **Response to Reviewer Lt5s (3/3)**
>
> >**Q3**. The authors may want to decompose some of technical details in Figure 1 into separate figures and move them closer to corresponding paragraphs.
>
> Thank you for your suggestion, which has greatly helped us improve the clarity of our paper. In the revised version, we have broken down some of the technical details in Figure 1 to enhance their comprehensibility.

---

> > ### Comment · Reviewer_Lt5s · 2024-11-24
> >
> > I have checked the revised manuscript and I would like to thank the authors for the efforts.

---

> ### Author Response · Authors · 2024-11-28
> **An Intuitive Comparison**
>
> Thank you again for your response and suggestions. We have identified a point that you might find interesting, which could better highlight the differences between our method and ZePT. Below, we provide a detailed comparison of the text descriptions used in ZePT and the attribute descriptions used in our approach.
>
> For a specific liver tumor patient,
> >**Text prompt used in ZePT:** "A liver tumor can arise in any of the liver's lobes or segments, and its precise location, whether in the right lobe, left lobe, caudate, or other segments, can be accurately depicted on abdominal CT scans. Liver tumors can manifest in various shapes, ranging from round to irregularly contoured masses. Anatomically, a liver tumor may appear as a hypoechoic or hyperechoic lesion on CT showing characteristics like enhancement patterns, presence of necrosis, and vascular invasion".
>
> >**Text prompt used in our method:** "Liver tumor. 1. Enhancement Status: enhanced CT; 2. Shape: Round-like; 3. Density: hypodense lesion; 4. density variations: Heterogeneous; 5. Surface Characteristics: ill-defined margin; 6. relationship with surrounding organs: no close relationship with adjacent organs; 7. specific features: presence of decreased density areas; 8. Location: Liver".
>
> As can be seen, ZePT uses the same general disease category knowledge for every patient, lacking the ability to learn patient-specific information. In contrast, our attribute descriptions provide fine-grained details about each patient's specific disease attributes. One of our key technical contributions, the multi-scale mask-attribute alignment, is designed to align disease region features with different attributes, forming multiple positive pairs for each lesion mask to establish fine-grained relationships between visual features and various disease attributes, thereby achieving better generalization performance. This approach goes beyond simple deep supervision.
>
> Aligning visual and language features has always been a key focus in vision-language pre-training research. Our approach is distinctive in aligning multiple diseases across different anatomical regions with distinct fundamental disease attributes. We hope this explanation addresses your concerns regarding our technical contributions.

---

> > ### Comment · Reviewer_Lt5s · 2024-12-02
> >
> > After careful consideration I increased my scores in light of attribute learning, a pre-LLM zero-shot learning idea which is neat and straightforward.

---

> > > ### Author Response · Authors · 2024-12-03
> > > **Response to Reviewer Lt5s**
> > >
> > > Thank you so much for recognizing our contributions. We are very grateful for your valuable advice, which has helped us think more deeply and has certainly improved our paper!

---

### Meta-Review · Area_Chair_Hfzn · 2024-12-23

**Metareview:**

This paper introduces Malenia, a framework for zero-shot lesion segmentation in 3D medical images, leveraging vision-language pre-training to align multi-scale mask representations with textual embeddings of lesion attributes. By decomposing radiological reports into attributes using LLMs and human annotators, the framework achieves fine-grained alignment between visual and textual features, enhancing its ability to generalize to unseen lesions. Through comprehensive experiments across three datasets and 12 lesion categories, Malenia demonstrates improved segmentation performance over existing zero-shot segmentation methods.

Strength: This approach to decomposing free-text reports into structured attributes reduces ambiguity and improves performance on unseen lesions. The framework addresses a critical need in clinical settings for handling diverse, emerging anomalies without labeled data. Empirical validation across multiple datasets highlights its robustness and effectiveness.

Weakness: Reviewers raise concerns about the method’s similarity with prior work, the privacy concerns by using GPT-4 for attribute extraction, lacking the extension to other imaging modalities and how to handle ambiguous lesion boundaries. The author has addressed these concerns well in the rebuttal period.

Considering the paper’s contribution based on the promising performance as well as the improved revision during rebuttal, I would lean to accept.

**Additional Comments On Reviewer Discussion:**

During the rebuttal, the author provided a detailed response to all reviewers’ comments, which has well addressed most of the concerns. Most reviewers have responded to the rebuttal and two reviewers have raised the score to reflect the paper improvement from the author’s response and manuscript revision. After the rebuttal, the scores converged to above borderline, while the only negative score comes from the reviewer who didn't respond to the rebuttal. After reading the paper, reviewers’ comments and rebuttals, I think the concerns have been addressed in a good manner.

---

### Decision · Program_Chairs · 2025-01-22

Accept (Poster)